# Technical note: Identification of two ice-nucleating regimes for dust-related cirrus clouds based on the relationship between number concentrations of ice-nucleating particles and ice crystals

Yun He[1,2,3,*], Zhenping Yin[4,*], Fuchao Liu [1,2,3], Fan Yi[1,2,3]

[1]School of Electronic Information, Wuhan University, Wuhan 430072, China
[2] Key Laboratory of Geospace Environment and Geodesy, Ministry of Education, Wuhan 430072, China
[3]State Observatory for Atmospheric Remote Sensing, Wuhan 430072, China.
[4]School of Remote Sensing and Information Engineering, Wuhan University, Wuhan 430072, China.

*Correspondence to*: Yun He (heyun@whu.edu.cn), Zhenping Yin (zp.yin@whu.edu.cn)

**Abstract.** Large amounts of dust aerosols are lifted to the upper troposphere every year and play a major role in cirrus formation by acting as efficient ice nuclei. However, the relative importance of heterogeneous nucleation and spontaneous homogenous nucleation in dust-related cirrus clouds is still not well evaluated globally. Here, based on space-borne observations, we propose a method to identify two ice-nucleating regimes of dust-related cirrus clouds, i.e., (1) sole presence of heterogeneous nucleation and (2) competition between heterogeneous and homogeneous nucleation, by characterizing the relationship between dust ice-nucleating particle concentrations (INPC) calculated from the Cloud-Aerosol LIdar with Orthogonal Polarization (CALIOP) using the POlarization LIdar PHOtometer Networking (POLIPHON) method and in-cloud ice crystal number concentration (ICNC) from the DARDAR (liDAR/raDAR) dataset. Two typical cirrus cases over central China are shown as a demonstration. In the first case, the upper part (near the cloud top) of a series of cirrus clouds successfully realized the INPC-ICNC closure, meaning that heterogeneous nucleation solely takes place; while the lower part of cirrus clouds showed the possible competition between heterogeneous and homogeneous nucleation. In the second case, the ICNCs in the cirrus cloud dramatically exceeded the dust INPCs in the vicinity by more than an order of magnitude, revealing that besides dust-induced heterogeneous nucleation, homogeneous nucleation also participated in ice formation and produced additional ice crystals. The proposed identification method is anticipated to apply in evaluating the influence of upper-troposphere dust on global cirrus formation and investigating the potential positive role of cirrus cloud thinning in the offset of climate warming.

## 1 Introduction

Cirrus clouds, composed of pure ice crystals, widely exist at the high altitude of the troposphere from ~5 km up to the tropopause, where the temperatures are usually below -38 ℃ (Heymsfield et al., 2017; Ge et al., 2018; Krämer et al., 2020). They approximately cover 20-25% of Earth's surface at any given time (Maloney et al., 2022), play a crucial role in climate

by altering the balance of solar and terrestrial radiation (IPCC, 2013), and potentially modulate water vapor balance between the upper troposphere and the lower stratosphere via convection (Jensen et al., 2013).

Cirrus clouds contribute a large uncertainty in general circulation models, resulting in an inadequate accuracy in predicting the rate and geographical pattern of climate change (Heymsfield et al., 2017). Due to the variety of their microphysical properties (particle size, number, and shape), it is even a challenge to convincingly draw a qualitative conclusion that cirrus

clouds cause either warming or cooling effect (Wolf et al., 2019). Cirrus clouds are typically classified into two categories in terms of forming mechanisms: in situ-origin cirrus and liquid-origin cirrus (Krämer et al., 2016, 2020). In situ-origin cirrus forms at colder altitudes (< -38 °C) accompanied by either fast updraft or slow updraft. In fast updraft situation, homogeneous freezing is the dominated freezing type since ice supersaturation quickly increases up to the homogeneous freezing threshold; in slow updraft situation, heterogeneous freezing (deposition freezing) is dominant first because ice supersaturation lies

between the heterogeneous and homogeneous freezing threshold (Krämer et al., 2009) and later homogeneous freezing is followed due to the depletion of ice-nucleating particles (INPs) and the persistence of cooling. Liquid-origin cirrus completely forms through heterogeneous freezing (immersion and contact freezing) and is then uplifted to the colder altitudes where homogeneous freezing can occur under high updraft conditions in addition to the heterogeneously formed ice crystals. Recently, numerous studies have reported that many types of aerosols existing at the upper troposphere (cirrus altitudes) can serve as

effective ice-nucleating particles (Cziczo et al., 2013), such as aviation soot (Tesche et al., 2016; Righi et al., 2021), volcanic aerosols (Sporre et al., 2022), wildfire smoke (Ansmann et al., 2021; Hu et al., 2022), dust aerosols (Kuebbeler et al., 2014), sulfate and nitrate particles (Che et al., 2021) and so on, making the in-situ heterogeneous formation at cirrus altitudes possible.

In consideration of the emission load and ice-nucleating efficiency, dust is the essential type among the upper-tropospheric INP particles. Abundant dust aerosols are elevated from the surface of desert areas by wind and convection every year and

then are mainly transported in the lower and middle troposphere (He et al., 2015, 2021a; Yin et al., 2021; Guo et al., 2017). Occasionally, a fraction of dust particles can be elevated to the upper troposphere by orographic uplift (Zhu et al., 2022), possibly initiating in-situ heterogeneous nucleation (Kanji et al., 2017; Froyd et al., 2022; Yang et al., 2022). Although homogeneous nucleation can be effective at temperatures of <-38 °C under suitable moisture conditions, heterogeneous nucleation may inevitably take place in cirrus clouds in the presence of dust particles (Ansmann et al., 2019a; He et al., 2022a).

Compared with homogeneous nucleation, the participation of heterogeneous nucleation strongly alters the microphysical properties of cirrus clouds, showing a reduction in ice crystal number concentration (ICNC) and a growth of crystal size. These optically thinner cirrus clouds absorb outgoing long-wave (LW) radiation from the surface and allow more LW radiation to emit to space, contributing to a cooling effect (Gasparini and Lohmann, 2016). This cooling effect prevails over the warming effect caused by the increased incoming solar radiation (warming), resulting in a net-positive radiative effect (cooling) on the

radiation budget of the Earth (Kuebbeler et al., 2014; Lohmann and Gasparini, 2017). As estimated by the simulation studies, the involvement of heterogeneous nucleation in cirrus formation would solely induce a net cloud forcing of -0.4 W·m$^{-2}$ (Liu et al., 2012), -2.0 W·m$^{-2}$ (Lohmann et al., 2008), or -0.94 W·m$^{-2}$ (Kuebbeler et al., 2014).

It is also worth noting that high ice supersaturation ratios of >1.4-1.5 are indispensable to triggering homogeneous freezing (Koop et al., 2000; Cziczo et al., 2013). Therefore, due to the supply of surrounding dust INPs, cirrus clouds (so-called dust-related cirrus clouds here) may form via heterogeneous nucleation solely, or alternatively, via the competition between heterogeneous and homogeneous ice nucleation (Zhao et al., 2019; Zhao et al., 2022), depending on both the temperature and ice supersaturation conditions. These two ice-nucleating mechanisms can result in quite discrepant microphysical properties in cirrus clouds. However, the competition between heterogeneous and homogeneous ice nucleation is still not well understood due to insufficient measurements (Spichtinger and Cziczo, 2010; Maloney et al., 2022; Kärcher et al., 2022). In a review report on cirrus clouds, Heymsfield et al. (2017) pointed out that measuring/documenting the relative importance of homogeneous versus heterogeneous nucleation of in-cloud ice crystals is considered one out of seven challenging tasks to be fulfilled in cirrus cloud research.

According to the abovementioned motivations, it is necessary to identify the heterogeneous-sole and competition ice-nucleating mechanisms of dust-related cirrus clouds. To realize this purpose, we propose to perform a closure study on the relationship between the dust INP concentration (INPC) and ICNC (Knopf et al., 2022) on the basis of the principle that one INP generates one ice crystal under the heterogeneous nucleation regime. It can be expected that the heterogeneous-sole case will successfully achieve the INPC-ICNC closure, while in the competition case, the in-cloud ICNC will further exceed the dust INPC attributed to the involvement of homogeneous freezing. Studies on the aerosol-cloud interaction by linking the INPC and ICNC were first successfully attempted based on airborne in situ observations (Prenni et al., 2009; Costa et al., 2017), and were extended to the ground-based active remote sensing approach (i.e., lidar-radar combinational observation) later benefiting from the development of retrieval techniques of ICNC (Bühl et al., 2019; Engelmann et al., 2021; Wieder et al., 2022) and INPC (Mamouri and Ansmann, 2014, 2015, 2016; Ansmann et al., 2019a).

To extend this method to a global scale, space-borne observations are necessary. Recently, the INPC-ICNC closure study was successfully conducted with active remote sensing from space, benefitting from the reliable ICNC values provided by DARDAR (liDAR/raDAR) product (Sourdeval et al., 2018; Gryspeerdt et al., 2018), which combines the observations of space-borne lidar and millimeter-wave radar. By comparing with in situ measurements, it has been proved that ICNC in cirrus clouds can be reliably represented by DARDAR product (Marinou et al., 2019; Krämer et al., 2020). Moreover, dust INPC can be retrieved from space-borne lidar observation with the POLIPHON (POlarization LIdar PHOtometer Networking) method. These two active remote-sensing instruments are able to accurately offer vertical-resolved information on clouds and aerosols, providing a unique way to study the dust-cirrus interaction. In consequence, in this study, we aim to identify the specific ice-nucleating mechanisms by comparing the ICNC in cirrus cloud provided by the DARDAR dataset with the dust INPC in the vicinity derived by POLIPHON method. Two typical cirrus cases over central China regions, where dust plumes frequently intrude during spring and winter (He et al., 2015, 2022c), are shown as a demonstration of the proposed method, which are favorable for validating some aspects of this method (such as the applicability of regional conversion factors in INP retrieval, the selection of optimal INP parameterization scheme, the confirmation of dust-related cirrus clouds, and so on) and conducting a robust long-term study on a global scale subsequently.

The organization of this paper is as follows. First, we briefly introduce the observational data and POLIPHON method used in the study. Then two typical case studies are presented to show the diversity of different ice-nucleating mechanisms. In the last section, some discussions, as well as conclusions, are given.

## 2 Data and methodology

### 2.1 CALIOP data

To obtain the profiles of aerosol optical properties, we used the observational data from a space-borne polarization lidar, i.e., the Cloud-Aerosol Lidar with Orthogonal Polarization (CALIOP). The CALIOP instrument was carried on the Cloud-Aerosol Lidar and Infrared Pathfinder Satellite Observation (CALIPSO) launched in April 2006 (Winker et al., 2007). CALIOP has three detecting channels and thus, can measure the elastic backscatters at both 532 nm and 1064 nm and the depolarization ratio at 532 nm. In this study, the level-2 aerosol profile data product (version 4.2) was used to provide the vertical distributions of the aerosol extinction coefficient, particle depolarization ratio, and atmospheric volume description (Omar et al., 2009). The atmospheric volume description product can further give information on the vertical feature mask (i.e., identifying cloud and aerosol), aerosol subtype, and cloud subtype.

'Cirrus' can be identified by cloud subtype product; 'dust' and 'polluted dust' can be identified by aerosol subtype product. Using these identifications together with the 532-nm total attenuated backscatter coefficient and volume polarization ratio (in CALIPSO level-1B data product), we can distinguish the dust-related cirrus clouds. The basic principle of case selection is that dust particles are observed closely in the vicinity (e.g., in vertical or horizontal directions) of cirrus cloud. More details about case selections can be found in He et al. (2022a).

Taking aerosol extinction coefficient $\alpha_p$, particle depolarization ratio $\delta_p$, and assumed aerosol lidar ratio into the calculation, we can obtain the dust backscatter coefficient $\beta_d$ (Tesche et al., 2009) and dust extinction coefficient $\alpha_d$ by using a constant dust lidar ratio of 45 sr (He et al., 2021a; Peng et al., 2021). The related calculation process is given in Table 1. Then, the dust extinction coefficient $\alpha_d$ will be taken as the input parameter in POLIPHON calculation.

### 2.2 ICNC derived from DARDAR dataset

To achieve the microphysical properties of cirrus clouds, we used the DARDAR dataset, which is the output of synergistic radar-lidar retrieval (Delanoë and Hogan, 2008, 2010). The DARDAR dataset retrieves the cloud properties by combining the measurements of the CALIOP instrument on CALIPSO satellite and the 94-GHz cloud profiling radar (CPR) on CloudSat satellite (launched in April 2006 together with CALIPSO). This lidar-radar combined approach is also broadly used in ground-based observation to retrieve cloud microphysical properties (Bühl et al., 2019; Wieder et al., 2022). Both satellites belong to NASA's 'A-train' constellation and thus can realize the collocated and quasi-simultaneous measurements of aerosols and clouds. It should be mentioned that nighttime measurements for CloudSat are only available until 2011.

In this study, the DARDAR-Cloud product is employed to provide the profiles of ice cloud properties, including the cloud extinction coefficient, cloud particle effective radius ($r_e$), and ice water content (IWC). The DARDAR-Nice profile product is used to obtain the profiles of ice crystal number concentration $n_{ice}$ within the clouds. The $n_{ice}$ values with ice-crystal diameters (i.e., maximum dimension) larger than 5 μm, 25 μm, and 100 μm are respectively derived with the approach presented by Sourdeval et al. (2018) and Gryspeerdt et al. (2018), using two parameters from the DARDAR-Cloud product, i.e., ice water content and normalization factor of the modified gamma size distribution ($N_0^*$). Both DARDAR-Cloud and DARDAR-Nice products have 60-m vertical and 1.7-km horizontal resolution.

The uncertainty in $n_{ice}$ is about 25% in lidar-radar condition and 50% in lidar- or radar-only conditions. Benefitting from the better transparent condition, $n_{ice}$ data product is considered more accurate for cirrus cloud than for mixed-phase cloud. In addition, high homogeneous nucleation rates would result in an additional 50% underestimation of $n_{ice}$ attributed to derivations from the assumed particle size distribution (Marinou et al., 2019). Therefore, Marinou et al. (2019) stated that the DARDAR-retrieved $n_{ice}$ can reflect the order of magnitude of the true $n_{ice}$. Moreover, Krämer et al. (2020) compared the $n_{ice}$ values from the DARDAR-Nice product with the in situ measuring results from five campaigns and found that there is an overestimation within a factor of 2 for $n_{ice}$ in DARDAR-Nice. They considered this offset for $n_{ice}$ tolerable given the variability of 6 orders of magnitude.

**2.3 INPC calculated by POLIPHON method**

The dust-cloud interaction is generally considered to take place if a cirrus cloud is embedded in a dust layer (Ansmann et al., 2019a; Marinou et al., 2019); in this case, the dust layer and cloud layer should have a spatial overlap either vertically or horizontally so that they can be considered as coupled. The dust-related INPC is estimated from the CALIOP observations in dust-existing regions in the vicinity of cirrus clouds. The dust INPC is calculated with the POLIPHON method using the dust extinction coefficient as the input parameter (Tesche et al., 2009; Mamouri and Ansmann, 2015, 2016, 2017), which is separated from the CALIOP-retrieved aerosol extinction coefficient with the one-step approach reported by Mamouri and Ansmann (2014). The calculation processes of the dust-related optical and ice-nucleating parameters as well as their estimated uncertainties are given in detail in Table 1. The meteorological parameters, including the pressure and temperature, are obtained from the radiosonde measurements (Guo et al., 2019, 2020, 2021) at a meteorological station (i.e., in Wuhan city (30.5°N,114.4°E)) in central China. It is assumed that the temperature and pressure profiles of the atmosphere between the weather station and the location CALIPSO observes are horizontally homogeneous.

The formation of cirrus clouds can be in situ-origin below -38 °C or liquid-origin from mixed-phase clouds above -38°C; thus, both immersion (nucleates from supercooled liquid droplet with INP immersed) and deposition (including water vapor deposits onto the insoluble surface of INP and pore condensation and freezing (PCF)) freezing modes may take place in ice nucleation in cirrus clouds (Kanji et al., 2017; Marcolli, 2014). Therefore, in the dust-INPC computation, we utilized the parameterization schemes of D10 (DeMott et al., 2010), D15 (DeMott et al., 2015), and U17-I (Ullrich et al., 2017) for immersion freezing, and U17-D (Ullrich et al., 2017) for deposition freezing. Besides, condensation freezing can be considered

a special type of immersion freezing; contacting freezing, which needs an INP to collide with a supercooled droplet, was ignored. Compared with the S15 (Steinke et al., 2015) scheme, Marinou et al. (2019) found that the lidar-derived INPCs with the U17-D scheme are more coincident with those measured with unmanned aerial vehicles; thus, only the U17-D scheme was applied in calculating the dust-related INPC in deposition freezing mode. For the D10 and D15 parameterizations, the dust extinction coefficient should first be converted into the particle number concentration with a radius >250 nm ($n_{250,d}$) by multiplying by a conversion factor $c_{250,d}$. The retrieval of $c_{250,d}$ for the mixed dust situation (i.e., mix with local urban aerosols) over Wuhan has been introduced in detail by He et al. (2021b). In addition, another conversion factor $c_{s,d}$ is needed to convert the dust extinction coefficient into particle surface area concentration $S_d$, which is the input parameter for the U17 parameterization. Using the same sun photometer dataset for $c_{250,d}$ retrieval, by selecting mixed dust cases with a ground-based polarization lidar, we can obtain the $c_{s,d}$ of $1.99 \times 10^{-6}$ Mm m$^2$m$^{-3}$ for mixed dust situation (33 dust-intrusion days during 2011-2013) over Wuhan as shown in figure 1.

## 3 Observational results of two typical cirrus cloud cases

In principle, one INP can form one ice crystal via primary heterogeneous nucleation (Ansmann et al., 2019a); at cirrus altitudes, secondary ice production that usually occurs at modest supercooling (temperature ≥-10°C) can be excluded (Field et al., 2017). A closure study means that INPC and ICNC coincide with each other well, at least within the same order of magnitude, when ignoring collision and aggregation of ice crystals and taking the uncertainties in them into consideration (Ansmann et al., 2019; Marinou et al., 2019; Knopf et al., 2021). Here we give two typical case studies in detail. The first case realizes the closure between the dust INPC and the in-cloud ICNC, while the second case shows that the in-cloud ICNC dramatically exceeds the dust INPC. The evident differences between the two typical cases can be considered a good demonstration to reveal the dominant ice-nucleating mechanism in cirrus clouds.

### 3.1 Case on 15 May 2008: Sole presence of heterogeneous nucleation

Figure 2 shows the 532-nm total attenuated backscatter coefficient (TAB) and volume depolarization ratio $\delta_v$ obtained from the CALIOP Level-1B data product on 15 May 2008. In both black rectangles, when the footprint of the CALIPSO satellite passed over the region near Wuhan, three adjacent ice clouds with $\delta_v$ of >0.3 and intensive TAB (in red and gray) appeared at altitudes of approximately 9-11 km. These clouds were observed to be embedded in a dust plume with $\delta_v$ of 0.1-0.2 and relatively weaker TAB (in blue and yellow). The cloud top mainly overlapped with the top edges of the dust layer, indicating that dust particles may take part in heterogeneous ice formation in the cloud. Therefore, these ice clouds can be considered as dust-related cirrus clouds (He et al., 2021a, 2022a). As shown in figure 3, the dust plume and cirrus cloud are confirmed to occur at altitudes of 9-11 km by the atmospheric volume description data from the CALIOP level-2 aerosol profile product. Cirrus clouds and dust aerosols occurred in turns (figure 3a) in the latitude range of 32-35°N, revealing the probable dust-

cirrus interaction. The dust-cloud interaction is generally considered to take place if a cirrus cloud is embedded in a dust layer (Ansmann et al., 2019a; Marinou et al., 2019).

Figure 4 presents the ice cloud properties including the cloud extinction coefficient, cloud particle effective radius, and ice water content from the DARDAR-Cloud product on 15 May 2008 (the same time and location as figures 2 and 3). It is noticed that the cirrus clouds above ~9 km generally show a smaller extinction coefficient (<1.0 km$^{-1}$) and particle size (<50 μm) than those altocumulus and altostratus at lower altitudes. Taking the cirrus cloud at altitudes of 9-11 km and latitudes of 33.2-35.0°N into calculation, the in-cloud averaging extinction coefficient, cloud particle effective radius, and ice water content are 0.60 km$^{-1}$, 34.93 μm, and 13.89 mg m$^{-3}$, respectively. It should be mentioned that only the data points identified as cirrus clouds (with feature mask) and having valid data were used for calculating these averaging values.

The CALIOP profiles in the cloud-free regions nearby (32.0-33.2°N) are integrated to estimate the INPCs related to the cirrus cloud. The aerosol extinction coefficient and backscatter coefficient for total (dust + non-dust) and dust composition as well as particle depolarization ratio $\delta_\mathrm{p}$ are presented in figure 5. Above 9 km, two distinct dust layers appeared at 9.0-9.8 km and 10.6-10.9 km, respectively, both with a peak $\delta_\mathrm{p}$ of around 0.3. These dust layers also contained some non-dust components as seen from their contributions to the total extinction and backscatter coefficients (figures 5a and 5b), which can also be verified by the aerosol subtype of 'polluted dust' (in brown, see figure 3c). For the upper layer (10.6-10.9 km), the layer-averaged dust extinction coefficient was 13.0 Mm$^{-1}$; while for the lower-lying layer (at 9.0-9.8 km), the layer-averaged dust extinction coefficient was 13.6 Mm$^{-1}$. These dust layers were related to the heterogeneous ice nucleation in the cirrus cloud at altitudes of 9-11 km.

Figure 6 shows the concentration profiles of the dust mass, large particle number (with radius >250 nm), and surface area calculated from the dust extinction (see figure 5). The dust-related INPC profiles can be obtained in turn. At temperatures warmer than -35 ℃, immersion heterogeneous freezing was considered in INPC calculation with the parameterization schemes of D10, D15, and U17-I (immersion freezing for dust particles). Contacting freezing, which needs an INP to collide with a supercooled droplet, was less likely to occur and thus was ignored here (Hoffmann et al., 2013; Ansmann et al., 2019a). At temperatures colder than -35 ℃, the INPC for deposition freezing was calculated with the U17-D (for dust particles) parameterization which has an applicable temperature range of -33− -67 ℃. For the upper dust layer at altitudes of 10.6-10.9 km, the averaging dust-related INPCs are 0.405 L$^{-1}$ (0.042-0.931 L$^{-1}$) for U17-D with ice saturation ratio $S_\mathrm{i}$ of 1.15, 9.897 L$^{-1}$ (0.891-23.875 L$^{-1}$) for U17-D with $S_\mathrm{i}$ of 1.25, and 102.792 L$^{-1}$ (8.341-256.571 L$^{-1}$) for U17-D with $S_\mathrm{i}$ of 1.35. For the lower dust layer at altitudes of 9.0-9.8 km, the averaging dust-related INPCs are 0.003 L$^{-1}$ (0-0.007 L$^{-1}$) for U17-D with $S_\mathrm{i}$ of 1.15, 0.041 L$^{-1}$ (0.001-0.083 L$^{-1}$) for U17-D with $S_\mathrm{i}$ of 1.25, and 0.259 L$^{-1}$ (0.010-0.527 L$^{-1}$) for U17-D with $S_\mathrm{i}$ of 1.35.

The number concentrations of ice crystals larger than 5 μm, 25 μm, and 100 μm from the DARDAR-Nice product are shown in figure 6c. Here we denote these three types of ICNC values as $n_\mathrm{ice,5\mu m}$, $n_\mathrm{ice,25\mu m}$, and $n_\mathrm{ice,100\mu m}$, respectively. The averaging ICNCs within the upper part of cirrus clouds at altitudes of 10.6-10.9 km (corresponding to the lower dust layer) are 121.8 L$^{-1}$ (108.3-140.4 L$^{-1}$) for $n_\mathrm{ice,5\mu}$, 59.1 L$^{-1}$ (51.1-70.6 L$^{-1}$) for $n_\mathrm{ice,25\mu m}$, and 11.7 L$^{-1}$ (8.2-15.8 L$^{-1}$) for $n_\mathrm{ice,100\mu m}$,

respectively. It is of great interest to note that the INPCs retrieved by U17-D with $S_i$ of 1.35 are in good agreement with the in-cloud $n_{ice,5\mu}$ (ICNC / INPC ratio is 1.2) and $n_{ice,25\mu m}$ (ICNC / INPC ratio is 0.6) within this thin vertical extent (10.6-10.9 km) near the cloud top, where the initiation of ice formation usually takes place. While the INPC retrieved by U17-D with $S_i$ of 1.25 is closer to the in-cloud $n_{ice,100\mu m}$ (ICNC / INPC ratio is 1.2). These INPC-ICNC relationships show a good match within an order of magnitude, which thus can be considered successful closure. Moreover, the typical ICNC for heterogeneous freezing was reported to be 1-100 L$^{-1}$ by Cziczo et al. (2013) and 4.3-39 L$^{-1}$ by Ansmann et al. (2019a) (for the thin cirrus case therein), which are generally consistent with the observation in this case. Therefore, it can be concluded that heterogeneous nucleation solely occurred within the upper part of cirrus clouds, which is also explained by the plunge of ICNC at this altitude range.

For the lower part of cirrus clouds (9.0-9.8 km), the layer-averaging ICNCs are 186.8 L$^{-1}$ (129.8-233.1 L$^{-1}$) for $n_{ice,5\mu m}$, 86.7 L$^{-1}$ (60.8-106.7 L$^{-1}$) for $n_{ice,25\mu m}$, and 11.2 L$^{-1}$ (8.5-12.4 L$^{-1}$) for $n_{ice,100\mu m}$, respectively. The ICNCs at these altitudes are much larger than those within the upper part of clouds (10.6-10.9 km). The ICNC values are 2-3 orders of magnitude larger than the corresponding INPC values at the same altitudes. These large ICNCs are possibly attributed to the occurrence of homogeneous nucleation. Consequently, both heterogeneous and homogeneous nucleation might take place in this case. Without airborne in-situ observations, the process-level evolution of these cirrus clouds cannot be well described since space-borne active observations only provide snapshot information of clouds. As seen from the geometric shape (small horizontal coverage and large vertical extent) of cirrus clouds, they were likely to form via homogeneous nucleation first accompanied by a fast updraft condition at lower altitudes, causing the large ICNCs at below. In this type of cirrus clouds, sedimentation of ice crystals is considered not to play an important role. Then, along with the updraft, water vapor was consumed gradually and the in-cloud RH$_i$ would quickly reduce to close to saturation; thus, heterogeneous nucleation would take charge predominantly at higher altitudes (as discussed for the lower part of cirrus clouds in the last paragraph) (Krämer et al., 2016, 2020). Additionally, at warmer altitudes of 8-9 km (with temperatures of -27 $-$ -35 °C), successful closures were also realized for the immersion freezing mode, as seen from the relationships of $n_{ice,100\mu m}$$-$INPC (D10), $n_{ice,25\mu m}$$-$INPC(D15), and $n_{ice,5\mu m}$$-$ INPC(D15).

**3.2 Case on 31 December 2010: Competition between heterogeneous and homogeneous nucleation**

Another case was observed on 31 December 2010. Figure 7 shows the altitude-orbit contour plots of the 532-nm total attenuated backscatter coefficient and volume depolarization ratio $\delta_v$. The black rectangles mark a thick cirrus cloud extending from an altitude of 5 km to up to 10 km, with intense TAB (in gray and red) and enhanced $\delta_v$ of 0.2-0.4. Meanwhile, dust aerosols were observed in the vicinity of the cirrus cloud. The dust plume overall showed a wide horizontal extent within the latitudes of 22-33°N, presenting a descent trend from north to south along with the crossing path. For the cirrus cloud, a similar north-south height gradient was observed, suggesting the relationship between the cloud formation and dust particles.

Moreover, the presence of the cirrus cloud and dust plume can also be verified by the simultaneous cloud subtype and aerosol subtype data from the CALIOP Level-2 product (see figure 8).

The ice cloud properties, i.e., the cloud extinction coefficient, cloud particle effective radius, and ice water content, on 31 December 2010 provided by the DARDAR-Cloud product are represented in figure 9. The time and location for the measurement are the same as those in figures 7 and 8. Taking the cirrus cloud at altitudes of 5-10 km and latitudes of 33-35°N into calculation, the in-cloud average extinction coefficient, cloud particle effective radius, and ice water content are 0.47 km$^{-1}$, 45.61 μm, and 14.10 mg m$^{-3}$, respectively. Note that only the data points identified as cirrus clouds (with feature mask) and having valid data were used for calculating these averaging values.

The cloud-free CALIOP profiles in central China (31.3-32.5°N) are integrated to estimate the dust extinction coefficient and in turn, the INPC near the cirrus cloud. Dust particles contained in the dust plume may undergo sedimentation (especially for coarse-mode dust particles) to a certain extent during the long-range transport, causing a variation in dust number concentration within the dust plume. The time scale for cirrus cloud formation is much smaller than that for dust transport (generally several tens of hours or a couple of days), suggesting that the removal of dust particles is negligible during cirrus formation. Therefore, it can be reasonably assumed that the dust number concentration contained in the cirrus cloud is comparable to that in the adjacent dust layer. Figure 10 shows the aerosol extinction coefficient and backscatter coefficient for total (dust + non-dust) and dust composition, and the particle depolarization ratio $\delta_p$. Two distinct dust layers can be seen clearly. The upper dust layer was located at altitudes of 8.5-9.6 km with a peak $\delta_p$ exceeding 0.3, revealing the presence of pure dust particles. Within this dust layer, dust extinction contributed the most proportion of total aerosol extinction; the layer-averaged dust extinction coefficient reached up to 21.2 Mm$^{-1}$, with a peak value of 56.9 Mm$^{-1}$ at around 8.9 km. As for the lower-lying dust layer at altitudes of 5.0-6.8 km, the layer-averaged dust extinction coefficient was 12.3 Mm$^{-1}$, nearly a half compared with that for the upper layer; the maximum value was 23.9 Mm$^{-1}$ at an altitude of 5.7 km. Considering that the vertical extent of the cirrus cloud (5-10 km) and dust layers (i.e., 5.0-6.8 km and 8.5-9.6 km) are partly overlapped, dust particles possibly participated in the heterogeneous ice formation in the cirrus cloud (at least for the lower part of cloud), which will be further analyzed by comparing the dust-related INPC and ICNC within the cloud.

To examine the possible nucleation mechanism, the profiles of dust mass concentration, large particle (with radius >250 nm) number concentration, surface area, dust-related INPC, and in-cloud ICNC are calculated as presented in figure 11. For the lower-lying dust layer, the temperature lay in a warm range of >-25 °C, meaning that heterogeneous nucleation solely took place. To obtain the dust-related INPC, the parameterization schemes of D10, D15, and U17-I for immersion mode were taken into calculation. U17-I shows the best performance according to the comparison with $n_{ice,100\mu m}$. As for the upper dust layer, the temperature ranged from -35 °C to -40 °C; thus, the parameterization U17-D for deposition freezing was applied to compute the INPC. Ice saturation ratio values $S_i$ were assumed to be 1.15, 1.25, and 1.35, respectively. A larger ice saturation ratio resulted in more active INPs. For the upper dust layer at altitudes of 8.5-9.6 km, the averaging dust-related INPCs are 0.016

L$^{-1}$ (0.002-0.043 L$^{-1}$) for U17-D with $S_i$ of 1.15, 0.232 L$^{-1}$ (0.028-0.641 L$^{-1}$) for U17-D with $S_i$ of 1.25, and 1.667 L$^{-1}$ (0.202-4.672 L$^{-1}$) for U17-D with $S_i$ of 1.35.

The averaging ICNCs within the upper part of the cirrus cloud (corresponding to the upper dust layer at 8.5-9.6 km) are 93.4 L$^{-1}$ (59.8-129.7 L$^{-1}$) for $n_{ice,5\mu m}$, 44.2 L$^{-1}$ (27.5-62.1 L$^{-1}$) for $n_{ice,25\mu m}$, and 6.6 L$^{-1}$ (3.4-9.6 L$^{-1}$) for $n_{ice,100\mu m}$, respectively.
The maximum $n_{ice,5\mu m}$ reaches up to 129.7 L$^{-1}$. As reviewed by Heymsfield et al. (2017), the cirrus total ice concentrations generally fall in the range of 5-500 L$^{-1}$. However, based on a simulation study, Liu et al. (2012) found that the ICNCs in cirrus clouds are 100-300 L$^{-1}$ in the subtropical and mid-latitudes of the Northern Hemisphere, which is attributed to the coexistence of homogeneous and heterogeneous (associated with the Saharan and Asian dust) nucleation. They concluded that heterogeneous nucleation may deplete water vapor supply within the cloud parcel and further inhibit homogeneous nucleation,
resulting in a decreasing number of ice crystals. Compared with the typical ICNC of >300 L$^{-1}$ for pure homogeneous freezing, the in-cloud ICNCs are relatively smaller in this case, caused by the combination of homogeneous and heterogeneous nucleation (Kärcher et al., 2022).

The ICNCs containing smaller ice crystals (i.e., $n_{ice,5\mu}$ and $n_{ice,25\mu m}$) are more than an order of magnitude larger than the dust-related INPCs with $S_i$ of 1.35 (ICNC / INPC ratios are 55.9 and 26.5, respectively), indicating that a great number of
300 small-size ice crystals were formed via homogeneous nucleation (Liu et al., 2012; Cziczo et al., 2013). The $n_{ice,100\mu m}$ values are more comparable to these INPCs ($S_i = 1.35$) within an order of magnitude (ICNC / INPC ratio is 3.9), especially at ~9.5 km where $n_{ice,100\mu m}$ is substantially consistent with INPC, suggesting the participation of heterogeneous nucleation that usually produces ice crystals with large size and small number concentration (Cziczo et al., 2013). Therefore, we can conclude that both heterogeneous and homogeneous nucleation had taken place during the formation of this cirrus cloud. Haag et al.
(2003) reported that the combination of homogeneous and heterogeneous nucleation may probably take responsibility for the in-situ formation of cirrus clouds in the mid-latitude regions of the Northern Hemisphere. The in-situ measurements by DeMott et al. (2003) also supported the conclusion that cirrus formation can occur both by heterogeneous nucleation by insoluble particles (i.e., ice-nucleating particles) and homogeneous freezing of particles containing solutions. Since the observation of vertical velocity was lacking, it is hard to determine the exact process of cirrus formation. In this case, it is likely that the cirrus
cloud first formed via heterogeneous nucleation under a slow updraft condition and further switched to a 'second stage' in which homogeneous nucleation began to be dominant owing to the persistence of cooling/uplifting (Krämer et al., 2016). Krämer et al., (2016) mentioned that this type of cirrus usually has a large geographic coverage, which can also be seen in this case. Considering the large ICNC, homogeneous nucleation should be the dominant type of ice nucleation.

## 4 Discussions and conclusions

We propose a method to identify two ice-nucleating mechanisms of dust-related cirrus clouds based on space-borne observations (He et al., 2022a; Kärcher et al., 2022): (1) sole presence of heterogeneous nucleation; (2) competition between

heterogeneous and homogeneous nucleation. To distinguish the two mechanisms, the basic thought is to compare the ICNC within a cirrus cloud with the dust-related INPC in the vicinity. The in-cloud ICNC data are obtained from the DARDAR dataset (Sourdeval et al., 2018) upon the synergistic observations of CALIPSO and CloudSat satellites. The dust-related INPC is derived with the POLIPHON method using the observational data from the space-bore lidar CALIOP on CALIPSO satellite (Mamouri and Ansmann, 2014, 2015, 2016, 2017; Marinou et al., 2019; Ansmann et al., 2019b). We consider the deposition freezing and immersion freezing herein.

In this study, two typical cases corresponding to the abovementioned ice-nucleating mechanisms are studied in detail as a demonstration (see table 2). Both cases are observed in central China. The conversion factors obtained in Wuhan city (30.5°N, 114.4°E) are applied in POLIPHON calculation (He et al., 2021b). For the first case, the estimated INPC and ICNC values generally realize successful closure within the upper part of cirrus clouds, i.e., they are generally in good agreement within an order of magnitude (Ansmann et al., 2019a), indicating the ice formation rule that one ice-nucleating particle generates one ice crystal by heterogeneous nucleation. For the second case, the estimated ICNCs dramatically exceed the INPCs for more than an order of magnitude, which can even reach up to several orders of magnitude, meaning that homogeneous nucleation is involved in ice formation and additionally produces a surging number of ice crystals.

A conceptual sketch of two ice-nucleating mechanisms is shown in figure 12. The heterogeneous-sole situation is generally composed of fewer ice crystals with large size, allowing more solar radiation to enter the atmosphere and emitting more LW radiation back to space. In contrast, the competition situation probably leads to an optically denser cirrus cloud containing numerous smaller ice crystals produced by homogeneous nucleation, reflecting more solar radiation to space as well as retaining more LW radiation in the atmosphere (i.e., emitting less LW radiation to space) (DeMott et al., 2010; Kuebbeler et al., 2014).

Considering the integration of incoming solar radiation and outgoing LW radiation, unseeded (homogeneously-formed) cirrus clouds are traditionally considered to result in a net warming effect on Earth's atmosphere (5.7 W·m$^{-2}$ as given by Gasparini and Lohmann, 2016). However, the diversity of cirrus-formation regimes induces a vital source of uncertainties in Earth's radiation budget as well as in weather and climate predictions (Kienast-Sjögren et al., 2016). Spichtinger and Cziczo (2009) mentioned that both net warming induced by thin cirrus clouds and net cooling by thick cirrus clouds are possible to occur. Fusina et al. (2007) reported that under certain conditions ice crystal number concentrations play a crucial role in the transition between net warming and cooling. In recent years, there has been rising a new concept, i.e., 'cirrus cloud thinning', in light of the assumption that more cirrus clouds nucleate via heterogeneous freezing. It is even under discussion that cirrus clouds thinning can offset current climate warming (Lohmann and Gasparini, 2017). The proposed method herein may be conducive to understanding the potential effect of cirrus cloud thinning.

By comparing the ICNC and INPC, this study presents a demonstration of ice-nucleating regimes identification for the dust-related cirrus clouds. However, there are still many efforts needed to pay. In the future, a long-term study will be conducted with the high-quality DARDAR product during 2006-2010, when the simultaneous nighttime observations of CloudSat and CALIPSO are available. In addition, we selected the central China region to provide an exemplary observation; benefiting

from the space-borne observations, such a study is expected to extend to a global scale so that it can potentially offer constructive suggestions of cirrus representation to the current climate models (Froyd et al., 2022; Maloney et al., 2022). To realize this goal, regional POLIPHON conversion factors shall be retrieved using sun photometer data from not only global AERONET (Holben et al., 1998; Ansmann et al., 2019b) but also some other regional networks, such as SONET (Sun-sky radiometer Observation NETwork, Li et al., 2018), CARSNET (China Aerosol Remote Sensing Network, Che et al., 2009), and SKYNET (Sky Radiometer Network, Nakajima et al., 2007), to cover more diverse aerosol types and complete aerosol optical/microphysical properties along the long-range transport path. Moreover, the proposed method using the measurement data from space is based on the 'snapshot' observation; to validate its reliability and realize a process-level observation, intercomparison with simultaneous ground-based observations using the same approach as well as with in-situ measurements conducted by aircraft are also necessary (Bühl et al., 2019; Engelmann et al., 2021; Wieder et al., 2022). CALIOP level-2 data product with the 5-km horizontal resolution cannot satisfy the accurate identification of dust layer and cirrus cloud on a small scale (Vaillant de Guélis et al. 2022), causing a potential to overestimate dust-related INPC, which can be solved by ground-based lidar observations with higher spatio-temporal resolution. With ground-based observations, the involved measurements of the Doppler velocity of ice crystals and the vertical velocity of airflows will be more beneficial to determine the accurate ICNC and the process-level characterization of cirrus formation (Bühl et al., 2015, 2016, 2019; Radenz et al., 2018, 2021). In addition, the future launch of the EarthCARE satellite can promote our understanding of cloud processes (Illingworth et al., 2015), since its 94.05-GHz cloud profiling radar can possess the capability of Doppler detection so that the in-cloud ICNC will be determined more accurately under the better constraint of the ice-particle size spectrum. These efforts will improve our inadequate understanding of the impact of upper-troposphere dust on global climate (Yang et al., 2022; Zhu et al., 2022).

**Data availability**

Sun photometer data used to generate the results of this paper are available at the website (doi:10.5281/zenodo.4683015). CALIPSO data used in this work can be accessed through the website https://subset.larc.nasa.gov. Radiosonde data can be obtained at the following website http://weather.uwyo.edu/upperair/sounding.html. DARDAR products are accessible at the website https://www.icare.univ-lille.fr.

**Author Contributions**

Yun He conceived the research, analyzed the data, acquired the research funding, and wrote the manuscript. Zhenping Yin analyzed the data, participated in the scientific discussions, and reviewed and proofread the manuscript. Fuchao Liu reviewed and proofread the manuscript. Fan Yi acquired the research funding and led the study.

## Competing interests

The authors declare that they have no conflict of interest.

## Acknowledgments

This work is supported by the National Natural Science Foundation of China (grant nos. 42005101 and 41927804), the Hubei Provincial Natural Science Foundation of China (grant no. 2020CFB229), the Fundamental Research Funds for the Central Universities (grant nos. 2042020kf0018 and 2042021kf1066), and the Meridian Space Weather Monitoring Project (China).

The authors thank the University of Wyoming for providing the radiosonde data, the Atmospheric Science Data Central (ASDC) at the NASA Langley Research Center for providing the CALIPSO data, the ICARE Thematic Center for generating and storing the DARDAR products.

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

**Table 1. Calculations for dust-related optical and ice-nucleating parameters (Tesche et al., 2009; Marinou et al., 2019; DeMott et al., 2010, 2015; Steinke et al., 2015; Ullrich et al., 2017). The estimated uncertainties in these parameters are also given (Ansmann et al., 2019b). The applicative temperature ranges are -30 to -14 °C for U17-I(d), -67 to -33 °C for U17-D(d) (Ullrich et al., 2017), -35 to -21 °C for D15 (DeMott et al., 2015), and -35 to -9 °C for D10 (DeMott et al., 2010), respectively. The profiles of temperature $T(z)$ (in Kelvin) and pressure $p(z)$ are provided by the most recently launched radiosonde in Wuhan; $p_0$ and $T_0$ are the standard pressure**
**and temperature (Mamouri and Ansmann, 2015). $n_s$ denotes the ice nucleation active site (INAS) density needed in the U17 parameterization scheme (Ullrich et al., 2017). $S_i$ denotes the ice saturation ratio.**

| Dust-related parameters | Computation | Uncertainty | References |
|---|---|---|---|
| Dust backscatter $\beta_d$ (Mm$^{-1}$ sr$^{-1}$) | $$\beta_d(z) = \beta_p(z)\frac{(\delta_p(z) - \delta_{nd})(1 + \delta_d)}{(\delta_d - \delta_{nd})\left(1 + \delta_p(z)\right)}$$ $$\delta_d = 0.31, \delta_{nd} = 0.05$$ | 10-30% | (Tesche et al., 2009) |
| Dust extinction $\alpha_d$ (Mm$^{-1}$) | $$\alpha_d(z) = LR_d \times \beta_d(z)$$ | 15-25% | (Ansmann et al., 2019b) |
| Dust mass conc. $M_d$ (µg m$^{-3}$) | $$M_d(z) = \rho_d \times \alpha_d(z) \times c_{v,d}$$ $$\rho_d = 2.6 \text{ g cm}^{-3}$$ | 20-30% | (Ansmann et al., 2019b) |
| Particle number conc. (r> 250 nm) $n_{250,d}$ (cm$^{-3}$) | $$n_{250,d}(z) = \alpha_d(z) \times c_{250,d}$$ | 25-35% | (Ansmann et al., 2019b) |
| Particle surface conc. $S_d$ (m$^2$ cm$^{-3}$) | $$S_d(z) = \alpha_d(z) \times c_{s,d}$$ | 30-40% | (Ansmann et al., 2019b) |
| Particle surface conc. (r> 100 nm) $S_{100,d}$ (m$^2$ cm$^{-3}$) | $$S_{100,d}(z) = \alpha_d(z) \times c_{s,100,d}$$ | 20-30% | (Ansmann et al., 2019b) |
| D10 INP conc. $n_{INP}$ (L$^{-1}$) | $$n_{INP}(p_0,T_0,T(z)) = a \cdot (273.16 - T(z))^b \cdot n_{250,d}(p_0,T_0)^{[c(273.16-T(z))+d]}$$ $$n_{INP}(z) = n_{INP}(p_0,T_0,T(z)) \cdot (T_0 p(z))/(p_0 T(z))$$ $$a = 0.0000594; \, b = 3.33; \, c = 0.0265; \, d = 0.0033$$ | 50-500% | (DeMott et al., 2010) |
| D15 INP conc. $n_{INP}$ (L$^{-1}$) | $$n_{INP}(p_0,T_0,T(z)) = f_d \cdot n_{250,d}(p_0,T_0)^{[a_d(273.16-T(z))+b_d]} \cdot \exp\left[c_d(273.16-T(z)) + d_d\right]$$ $$n_{INP}(z) = n_{INP}(p_0,T_0,T(z)) \cdot (T_0 p(z))/p_0(T(z))$$ $$a_d = 0; \, b_d = 1.25; \, c_d = 0.46; \, d_d = -11.6; \, f_d = 3.0$$ | 50-500% | (DeMott et al., 2015) |
| U17-I(d) INP conc. $n_{INP}$(L$^{-1}$) | $$n_{INP}(z) = S_d(z) \times n_s(T(z))$$ $$n_s(T(z)) = \exp[150.577 - 0.517 \cdot T(z)]$$ | 50-500% | (Ullrich et al., 2017) |
| U17-D(d) INP conc. $n_{INP}$ (L$^{-1}$) | $$n_{INP}(z) = S_d(z) \times n_s(T(z),S_i)$$ $$n_s(T(z),S_i) = \exp\{a_u(S_i - 1)^{1/4} \cos[b_u(T(z) - c_u)]^2 \cot^{-1}[d_u(T(z) - e_u)]/\pi\}$$ $$a_u = 285.692; \, b_u = 0.017; \, c_u = 256.692; \, d_u = 0.08; \, e_u = 200.745$$ | 50-500% | (Ullrich et al., 2017) |
| S15 INP conc. $n_{INP}$ (L$^{-1}$) | $$n_{INP}(z) = S_d(z) \times n_s(T(z))$$ $$n_s(T(z)) = 1.88 \times 10^5 \cdot \exp[0.2659 \cdot \chi(T(z),S_i)]$$ $$\chi(T(z),S_i) = -(T(z) - 273.2) + (S_i - 1) \times 100$$ | 50-500% | (Steinke et al., 2015) |

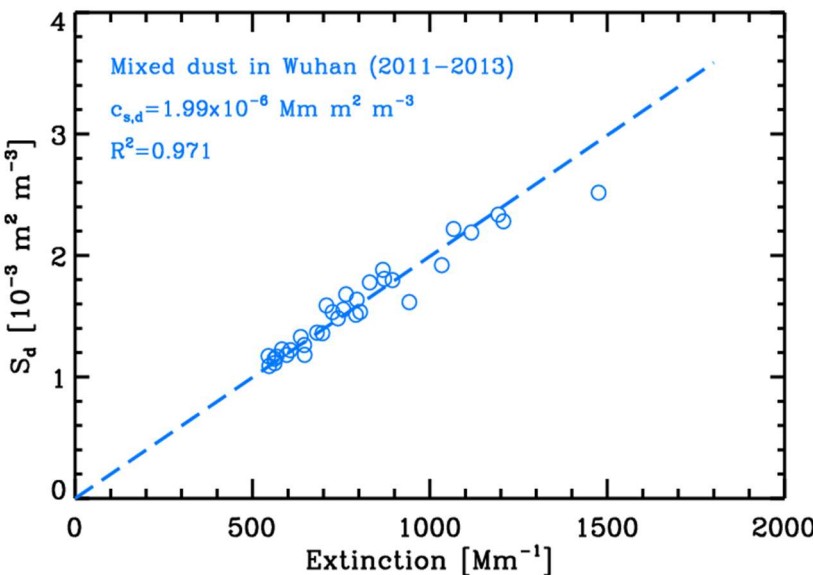

**Figure 1. Relationship between extinction coefficient and surface area concentration $S_d$ for mixed dust in Wuhan. Correlations are given by sun photometer observations during dust-intrusion days that are verified by ground-based polarization lidar (He et al., 2021b). Each hollow circle denotes a pair of daily averaged values for the dust occurrence period of a dust-intrusion day. The slope of the dashed blue line is defined as the extinction-to-surface area conversion factor (Ansmann et al., 2019b). All the circles are obtained from the dust-intrusion days during 2011–2013.**


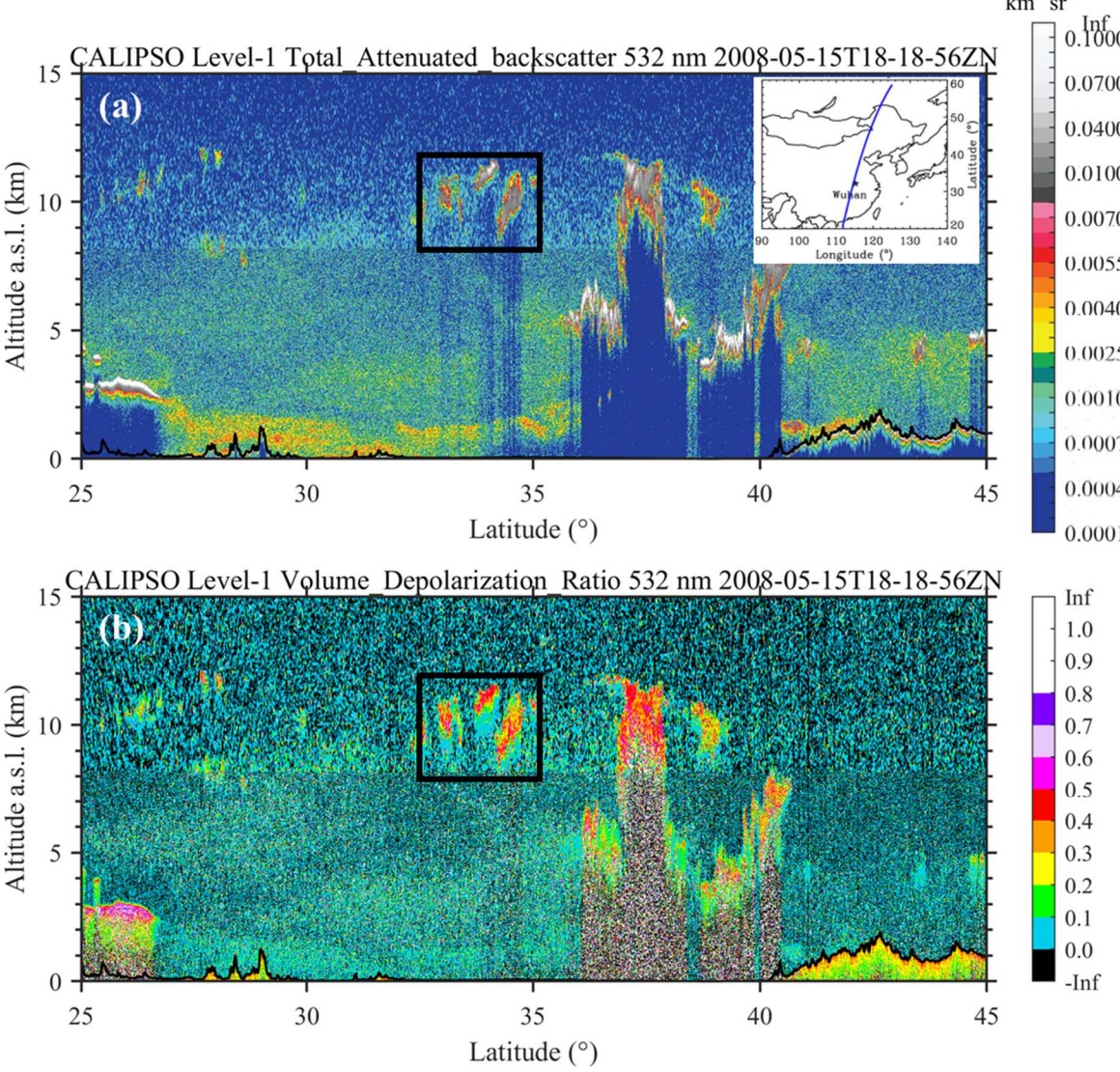

**Figure 2. CALIPSO altitude-orbit cross section measurements of the CALIOP Level-1B 532-nm (a) total attenuated backscatter coefficient and (b) volume depolarization ratio product on 15 May 2008. The corresponding orbit is 2008-05-15T18-18-56ZN. The black rectangles mark the dust-related cirrus cloud.**


**Figure 3. CALIPSO altitude-orbit cross section measurements of the CALIOP Level-2 (a) vertical feature mask, (b) cloud subtype, and (c) aerosol subtype product on 15 May 2008. The corresponding orbit is 2008-05-15T18-18-56ZN. The footprint of the satellite is the same as shown in figure 2.**


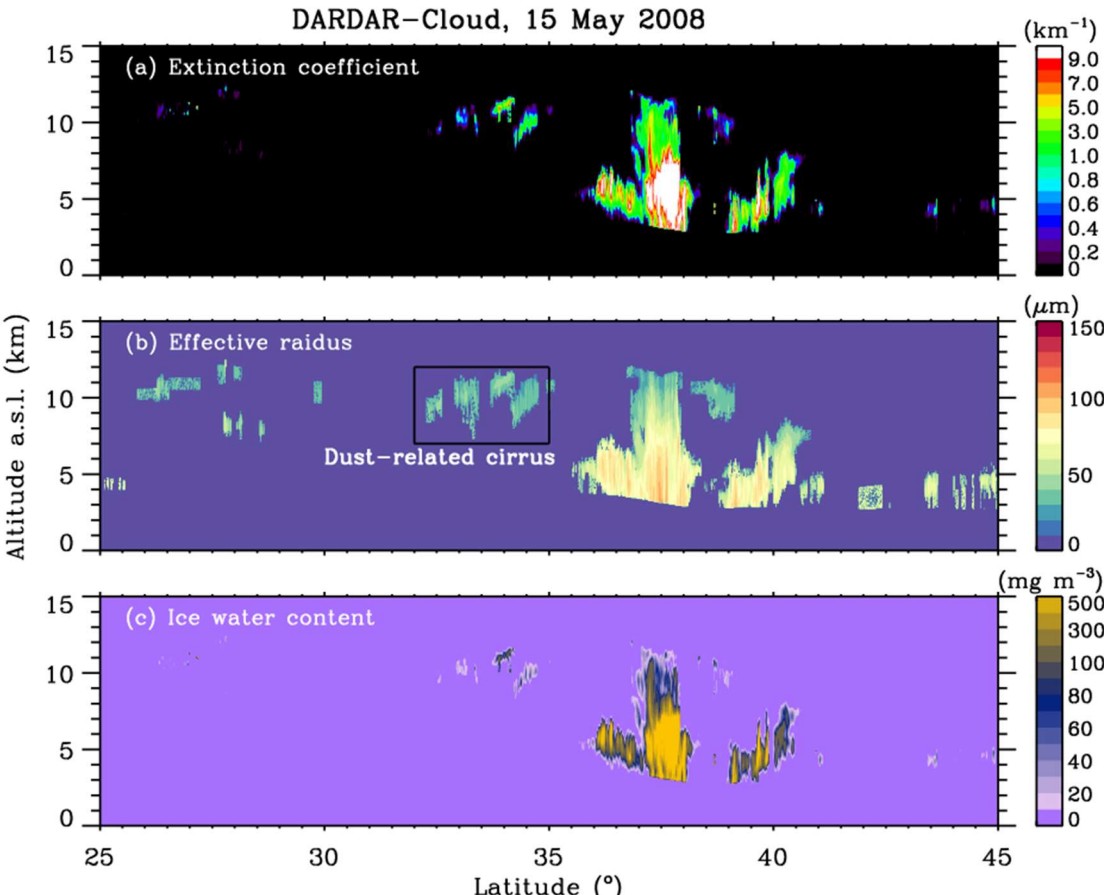

**Figure 4.** Altitude-orbit cross section of the (a) cloud extinction coefficient, (b) cloud particle effective radius, and (c) ice water content from the DARDAR-Cloud product on 15 May 2008. The footprint of the satellite is the same as shown in figure 2.


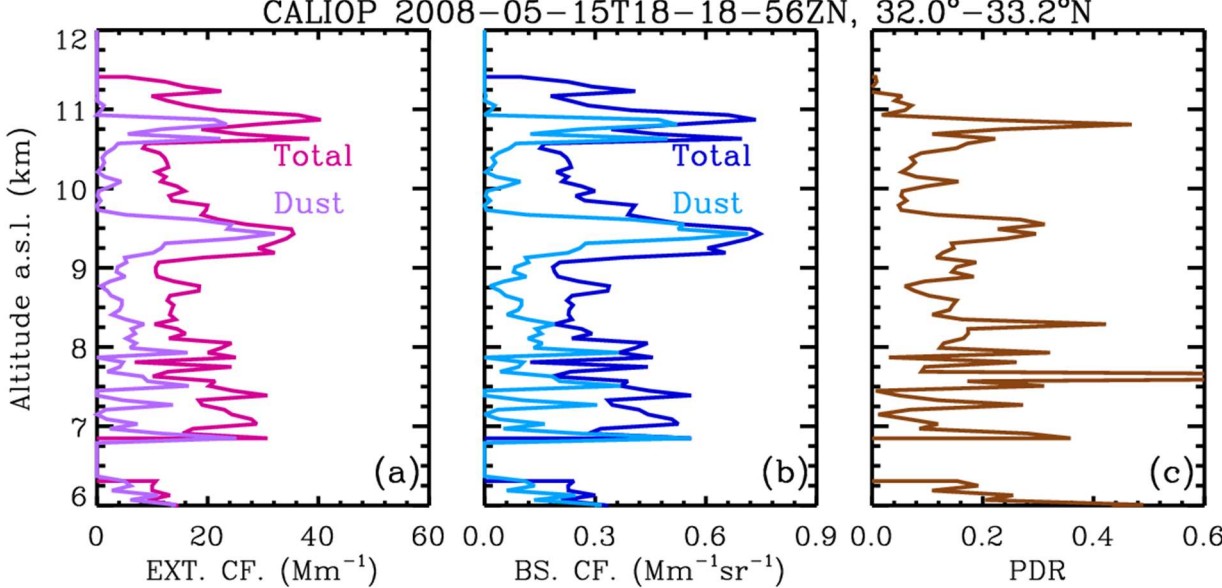

**Figure 5.** Profiles of the 532-nm (a) dust and total (dust + non-dust) extinction coefficient, (b) dust and total (dust + non-dust) backscatter coefficient, and (c) particle depolarization ratio obtained/calculated from the CALIOP Level-2 aerosol profile product on 15 May 2008. The profiles within the latitude range of 32.0-33.2°N are integrated here.


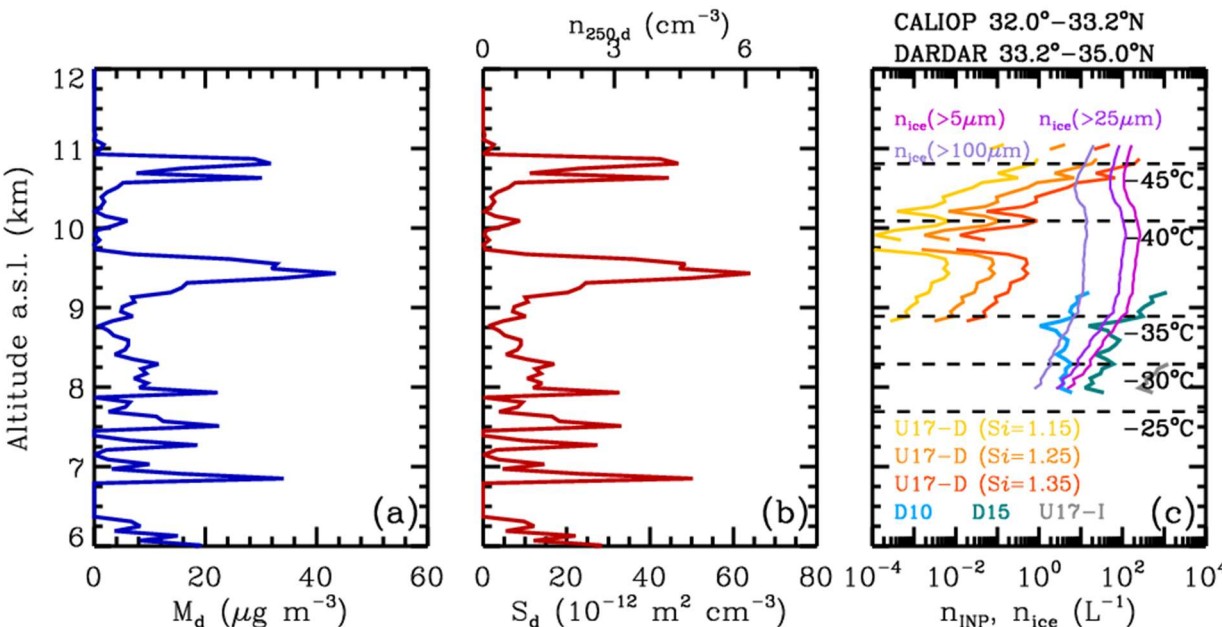

**Figure 6.** Profiles of the (a) dust mass concentration $M_d$, (b) particle number concentration (with radius >250 nm) $n_{250,d}$ and surface area concentration $S_d$, and (c) ice-nucleating particle concentration $n_{INP}$ (derived by the POLIPHON method using the parameterization schemes of D10, D15, U17-D, and U17-I) and ice crystal number concentration $n_{ice}$ (from DARDAR-Nice data product) on 15 May 2008. INP-related parameters are calculated from the dust-related optical properties given in figure 5 (corresponding to the CALIOP footprints between 32.0-33.2°N). For the DARDAR-Nice product, the profiles within the latitude range of 33.2-35.0°N are integrated. $S_i$ denotes the ice saturation ratio.

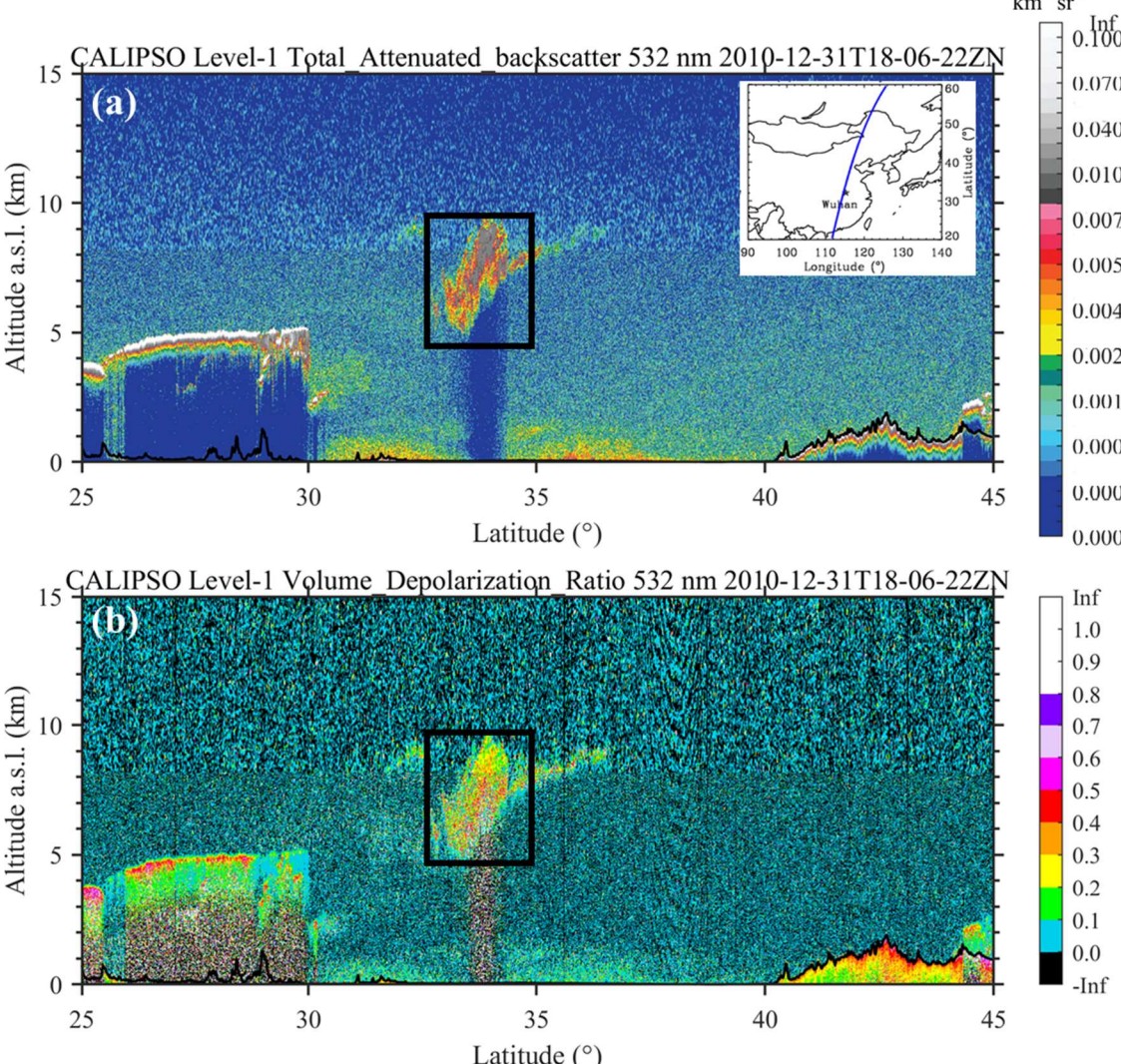

**Figure 7. CALIPSO altitude-orbit cross section measurements of the CALIOP Level-1B 532-nm (a) total attenuated backscatter coefficient and (b) volume depolarization ratio product on 31 December 2010. The corresponding orbit is 2010-12-31T18-06-22ZN. The black rectangles mark the dust-related cirrus cloud.**

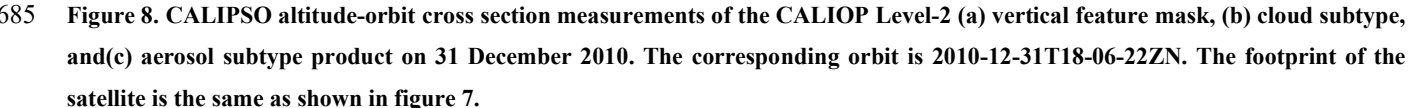

**Figure 8. CALIPSO altitude-orbit cross section measurements of the CALIOP Level-2 (a) vertical feature mask, (b) cloud subtype, and(c) aerosol subtype product on 31 December 2010. The corresponding orbit is 2010-12-31T18-06-22ZN. The footprint of the satellite is the same as shown in figure 7.**

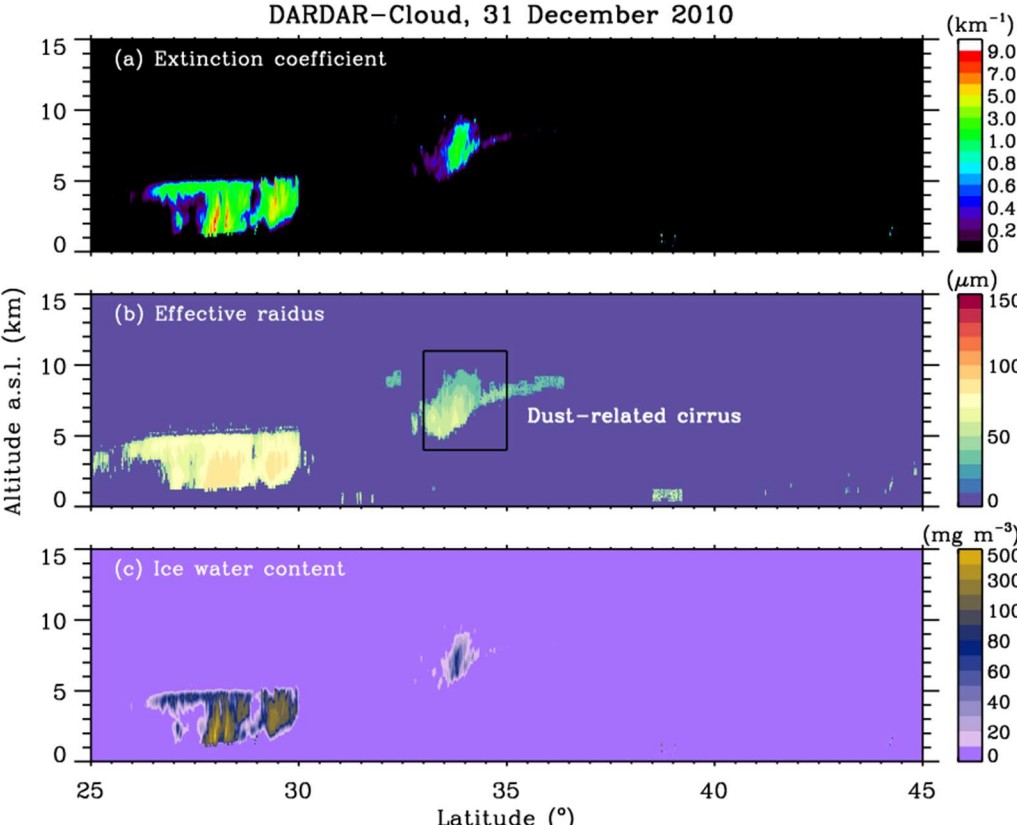

**Figure 9. Altitude-orbit cross section of the (a) cloud extinction coefficient, (b) cloud particle effective radius, and (c) ice water content from the DARDAR-Cloud product on 31 December 2010. The footprint of the satellite is the same as shown in figure 7.**

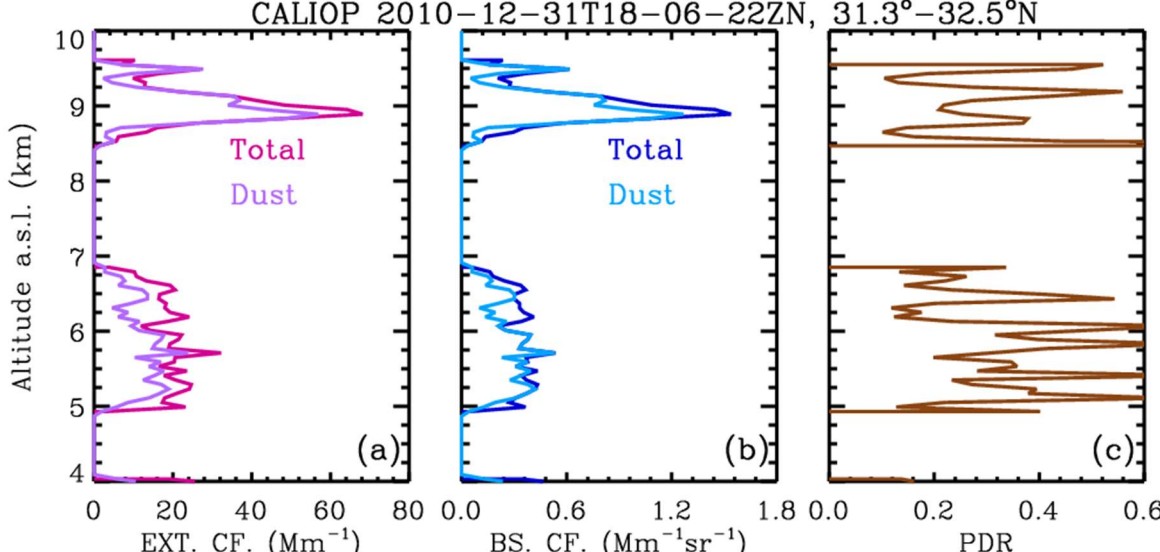

**Figure 10.** Profiles of the 532-nm (a) dust and total (dust + non-dust) extinction coefficient, (b) dust and total (dust + non-dust) backscatter coefficient, and (c) particle depolarization ratio obtained/calculated from the CALIOP Level-2 aerosol profile product on 31 December 2010. The profiles within the latitude range of 31.3-32.5°N are integrated here.

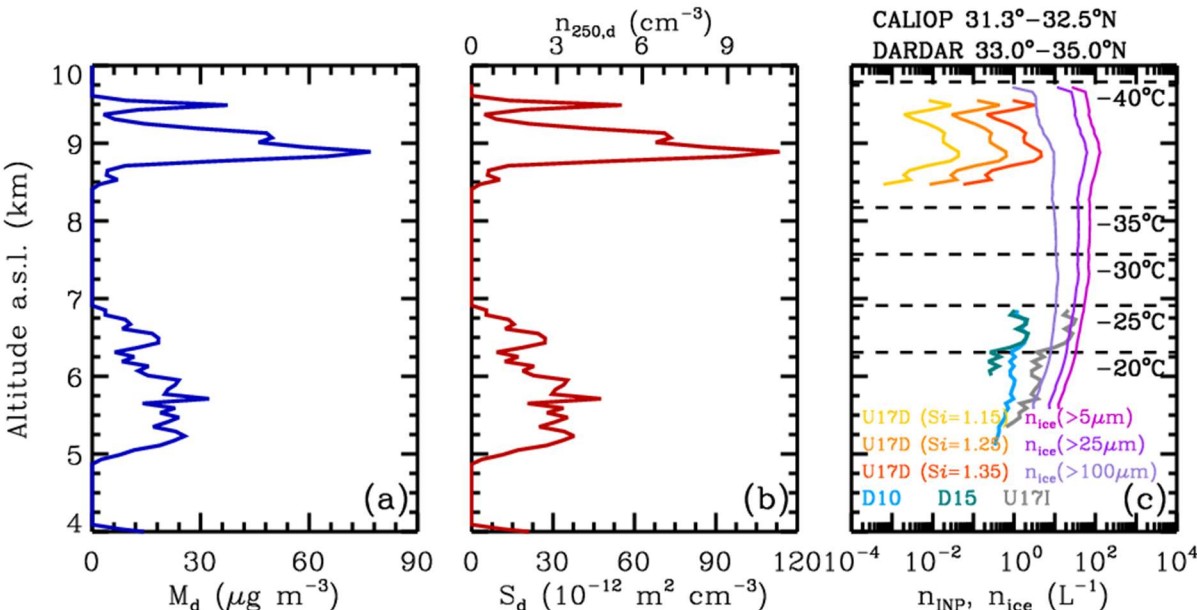

**Figure 11. Profiles of the (a) dust mass concentration $M_d$, (b) particle number concentration (with radius >250 nm) $n_{250,d}$ and surface area concentration $S_d$, and (c) ice-nucleating particle concentration $n_{INP}$ (derived by the POLIPHON method using the parameterization schemes of D10, D15, U17-D, and U17-I) and ice crystal number concentration $n_{ice}$ (from DARDAR-Nice data product) on 31 December 2010. INP-related parameters are calculated from the dust-related optical properties given in figure 10 (corresponding to the CALIOP footprints between 31.3-32.5°N). For the DARDAR-Nice product, the profiles within the latitude range of 33.0-35.0°N are integrated. $S_i$ denotes the ice saturation ratio.**

**Table 2. Overview of in-cloud ICNC and dust INPC for the two cases on 15 May 2008 and 31 December 2010 in central China regions. The ICNC values are provided by the DARDAR Nice data product. The INPC values are retrieved from CALIOP dust extinction coefficient based on the POLIPHON method. The layer-averaging values for ICNC and INPC are given together with the minimum and maximum values in parentheses. $S_i$ denotes the ice saturation ratio.**

| Parameter | 15 May 2008 | | 31 December 2010 |
|---|---|---|---|
| Ice-nucleating type | Heterogeneous-sole nucleation | Competition between heterogeneous and homogeneous nucleation | Competition between heterogeneous and homogeneous nucleation |
| Altitude range of cirrus (km) | 10.6 – 10.9 (upper part of the cirrus) | 9.0 – 9.8 (lower part of the cirrus) | 8.5 – 9.6 (upper part of the cirrus) |
| Latitude range of cirrus (°N) | 33.2 – 35.0 | 33.2 – 35.0 | 33.0 – 35.0 |
| Temperature range of cloud (°C) | -40 – -45 | -35 – -40 | -35 – -40 |
| ICNC, $n_{ice,5\mu m}$ (L$^{-1}$) | 121.8 (108.3 – 140.4) | 186.8 (129.8 – 233.1) | 93.4 (59.8 – 129.7) |
| ICNC, $n_{ice,25\mu m}$ (L$^{-1}$) | 59.5 (51.1 – 70.6) | 86.7 (60.8 – 106.8) | 44.2 (27.5 – 62.1) |
| ICNC, $n_{ice,100\mu m}$ (L$^{-1}$) | 11.6 (8.2 – 15.8) | 11.2 (8.5 – 12.4) | 6.6 (3.4 – 9.6) |
| Altitude range of dust layer (km) | 10.6 – 10.9 | 9.0 – 9.8 | 8.5 – 9.6 |
| Latitude range of dust layer (°N) | 32.0 – 33.2 | 32.0 – 33.2 | 31.3 – 32.5 |
| Dust INP, U17-D, Si=1.15 (L$^{-1}$) | 0.405 (0.042 – 0.931) | 0.003 (0 – 0.007) | 0.016 (0.002 – 0.043) |
| Dust INP, U17-D, Si=1.25 (L$^{-1}$) | 9.897 (0.891 – 23.875) | 0.041 (0.001 – 0.083) | 0.232 (0.028 – 0.641) |
| Dust INP, U17-D, Si=1.35 (L$^{-1}$) | 102.792 (8.341 – 256.571) | 0.259 (0.010 – 0.527) | 1.667 (0.202 – 4.672) |


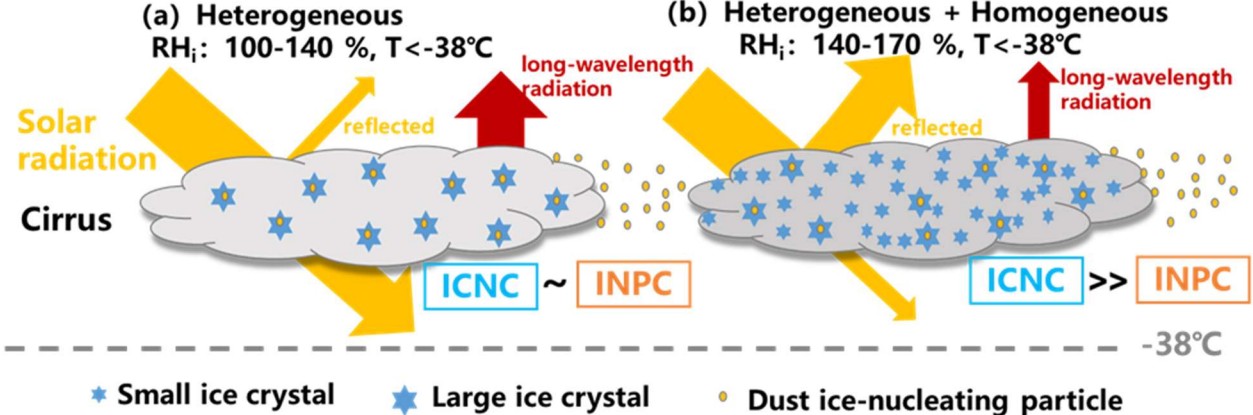

**Figure 12. Schematic illustration of the two ice-nucleating mechanisms of the dust-related cirrus clouds (i.e., cirrus occurs accompanied by dust particles ambient). (a) Heterogeneous nucleation solely takes place with the $RH_i$ of 100-140% and temperature of < -38℃. The ice crystals within the cirrus are relatively larger and the ICNC within the cirrus is comparable to the ambient dust INPC. This type of cirrus is always more transparent and allows more solar radiation to enter the middle and lower troposphere; while more long-wave radiation can be emitted to space. This 'seeded cirrus' results in a net cooling effect, partially offseting the warming effect of cirrus clouds on the Earth's atmosphere (Kuebbeler et al., 2014; Lohmann and Gasparini, 2017). (b) Competition between heterogeneous and homogeneous nucleation with the $RH_i$ of 140-170% and temperature of < -38 ℃. Besides the large ice crystals formed by heterogeneous nucleation, there are also numerous smaller ice crystals within the cirrus produced via homogeneous nucleation (Krämer et al., 2020). The ICNC within the cirrus is far beyond (one to several orders of magnitude) the ambient dust INPC (Froyd et al., 2022). This type of cirrus is optically denser and can reflect more solar radiation into space; however, less long-wave radiation can be emitted to space.**

