# Peer review of "Technical note: Identification of two ice-nucleating regimes for dustrelated cirrus clouds based on the relationship between number concentrations of ice-nucleating particles and ice crystals"

_Atmospheric Chemistry and Physics, 2022_

## Referee Comment (RC2)

**Review of: "Technical note: Identification of two ice-nucleating regimes for dust- related cirrus clouds based on the relationship between number concentrations of ice-nucleating particles and ice crystals" by He et al.**

**Content**

This manuscript describes an approach to determine the ice crystal number concentration (ICNC) and dust related ice nucleation particle concentration (INPC) from satellite and sun photometer observations. Both quantities are compared to each other to examine the nucleation pathway (homogeneous or heterogeneous) of two single cirrus cloud cases in China.

**Overall impression and rating**

The overall impression of the manuscript is good in general. The manuscript is mostly written in a clear way and the most important aspects are considered. The presentation quality of the manuscript is besides small details good. It is well organized and the analysis and results are clearly structured and communicated. However, in some minor parts I cannot fully follow the argumentation. Especially, the explanation of the middle cloud part in case 1 where INPC and ICNC disagree is too short and insufficient. For these reasons, I recommend publication in ACP after some minor manuscript revisions.

**Specific comments/questions:**

- Page 2, lines 41-43, You are right that liquid origin clouds form completely heterogeneously, but at a later stage when reaching higher/colder altitudes additional homogeneous freezing can occur, if the updraft is fast enough. This can happen in convection and warm conveyor belts (WCB, see Kraemer et al. 2016). I recommend to add at the end of sentence: "...altitudes where homogeneous freezing can occur under high updraft conditions in addition to the heterogeneously formed ice crystals."
- Page 5, lines 137-138, What is actually meant by "interestingcirrus clouds" ? Are this cases where your ICNC-INPC closure worked or how do you select the "interesting cases". Maybe you can a little bit more specific in the text.
- Page 5, line 144, I find this sentence confusing. What is meant by "isotherm" in this context ? Cirrus clouds usually occur below -38 °C, but can also as completely frozen mixed-phase clouds above -38 °C. I recommend to rewrite this sentence to make it clearer.
- Page 6, line 168, The date in the headline does not fit to the case you are showing! (29 December 2010 and 15 May 2008). So please correct. Please also capitalize after the colon.
- Page 6, line 173, I do not understand the first part of the sentence: "Dust particles were full of the cloud-free regions". Can you please rephrase it!

- Page 6, lines 182-184, How do you determine the average values ? Just taking your bounding box shown in the figures or did you use the feature mask from Calipso ? Because of the very irregular shape of the cloud, this can have a large influence on the averaged values. The same comment applies for page 8 line 238-289.
- Page 8, lines 221-225, The explanation and discussion of the disagreement in the lower part of the cloud is definitely not sufficient and long enough. You argue with sedimentation of the heterogeneously formed ice crystals from the cloud top level, which can be an explanation. But if the heterogeneously formed ice crystals fall from above they also lower the ice concentration and also the INPC which are consumed by the formation process in the upper part of the cloud. In addition, new ice crystals in the upper part cannot form heterogeneously afterwards anymore because of low INPC values. Or you have to explain where new IN particles should come from. I also do not understand the argument with homogeneous freezing in the lower part of the cloud. Homogenous freezing would need higher vertical updrafts to maintain high supersaturations. Where should the higher vertical updrafts come from and why one could not find them in the top part of the cloud ? I also think that homogeneous freezing can still not be ruled out completely also for the upper part of the cloud especially with the argument of ice crystal sedimentation. I think this point should be discussed in in more detail in this Section of the paper.
- Page 9, lines 277-279, I cannot follow your conclusion that "heterogeneous nucleation would gradually be dominant" in a competition situation. When you uplift an air parcel you will increase relative humidity until a cloud is forming. Of course you would form a cloud heterogeneously first, but with further vertical updraft and thus cooling, relative humidity can increase again reaching homogeneous nucleation threshold even if you are consuming humidity by particle growth of heterogeneously formed ice crystals. Thus, forming heterogeneously and homogeneously exactly at the same time is not possible, but of course one after the other. And than I would be identify the dominance of formation mechanism by the ICNC. Given the high ICNC values in the upper part of the cloud, I would argue that you have a large dominance of homogeneous frozen ice crystals in the cirrus cloud. I suggest to rethink your conclusions and discussion in this point.
- Figure 12: In part (a) of the figure you write T<-38°C, while you write <0°C in the caption. In this case your argumentation about thin clouds (few large crystals) is only valid for in-situ cirrus clouds, I suggest to change the caption to T<-38°C as in the Figure. In part (b) the RHi values stated in the figure are not correct. In a competition case heterogeneous freezing still happens at RHi values 100-140%. Thus I suggest to change the values in the Figure and Caption to 100-170%.

**Technical comments/suggestions:**

- Page 1, line 27: Better write "~5km up to the tropopause".
- Page 2, line 32: Use the plural form "general circulation models".
- Page 3, line 84: I recommend to include Kraemer et al 2020 as reference in addition to Marinou et al. 2019. The authors also perform a comparison between in-situ and satellite ICNC.
- Page6, line 175: Please change "can be considered dust-related cirrus clouds" to "can be considered as dust-related cirrus clouds".
- Page 7, line 199: "above -35 °C isotherm". I guess you mean colder than -35 ? So please correct the wording.
- Page 7, lines 200-204, Table 2: Please explain what is meant by the parameter Si, because it is not mentioned in the text before or afterwards.
- Page 8, line 221: "ice crystals falling": Better use the common term "ice crystal sedimentation".
- Page 8, line 226: Please capitalize after the colon.

- Figure 2/3/4: I would recommend to zoom a little bit closer to your selected cirrus case by showing only date between e.g. 45-25° latitude. Than all features are better visible.
- Figure 3/4: Please use the same latitudinal projection as in Figure 2 to make the figures better comparable.
- Figure 8/9: Same comment for Figure 3/4 above.

**References**

Krämer, M., Rolf, C., Luebke, A., Afchine, A., Spelten, N., Costa, A., Meyer, J., Zöger, M., Smith, J., Herman, R. L., Buchholz, B., Ebert, V., Baumgardner, D., Borrmann, S., Klingebiel, M., and Avallone, L.: A microphysics guide to cirrus clouds – Part 1: Cirrus types, Atmos. Chem. Phys., 16, 3463–3483, https://doi.org/10.5194/acp-16-3463-2016, 2016.

---

## Author Comment (AC1)

**Response to reviewer #1**

**General Comments:**

This study investigates the relationship between number concentrations of ice-nucleating particles and ice crystals, based on which it proposed an identification method for two ice-nucleating regimes for dust-related cirrus clouds. The findings are interesting and worthy for publication after necessary revisions.

**Response:** We appreciate the reviewer's thoughtful review and constructive comments. All the comments have been addressed in the revised manuscript, and the responses to each comment are given below.

**Specific comments:**

**Comments 1:** *Line 40-41, considering the shattering of ice particles which can further serve as ice nuclei to help the heterogeneous formation of ice crystals, I am not sure if the description "and later homogeneous freezing is followed due to the depletion of ice-nucleating particles and the persistence of cooling" is accurate or not.*

**Response 1:** To our best knowledge, the shattering of ice particles indicates secondary ice production, including four types according to the current studies, i.e., rime splintering, collision fragmentation, droplet shattering, and sublimation fragmentation. These processes usually occur at modest supercooling temperature (>-10°C), as seen in a review from Field et al. (2017). However, cirrus clouds form at farther colder temperatures; thus, we consider the shattering of ice particles cannot serve as the additional ice nuclei in this 'second stage'. For in situ-origin cirrus formed in a slow updraft, Krämer et al. (2016) have stated that '*The formation mechanism of this cirrus type starts with HET freezing that is followed by a second, HOM ice nucleation event (since all IN are already consumed) if the cooling phase is long enough and temperature fluctuation do not cause a HOM freezing event earlier.' (Please see section 5.1.1 therein). As a result, we would like to retain this sentence in the revised manuscript.*

**Reference:**

Krämer, M., Rolf, C., Luebke, A., Afchine, A., Spelten, N., Costa, A., Meyer, J., Zöger, M., Smith, J., Herman, R. L., Buchholz, B., Ebert, V., Baumgardner, D., Borrmann, S., Klingebiel, M., and Avallone, L.: A microphysics guide to cirrus clouds – Part 1: Cirrus types, Atmos. Chem. Phys., 16, 3463-3483, doi.org/10.5194/acp-16-3463-2016, 2016.

Field, P., Lawson, P., Brown, G., Lloyd, C., Westbrook, D., Moisseev, A., Miltenberger, A., Nenes, A., Blyth, A., Choularton, T., Connolly, P., Bühl, J., Crosier, J., Cui, Z., Dearden, C., DeMott, P., Flossmann, A., Heymsfield, A., Huang, Y., Kalesse, H., Kanji, Z., Korolev, A., Kirchgaessner, A., Lasher-Trapp, S., Leisner, T., McFarquhar, G., Phillips, V., Stith, J., and Sullivan, S.: Secondary ice production – current state of the science and recommendations for the future, Meteorol. Mon., 58, 7.1–7.20, doi.org/10.1175/AMSMONOGRAPHS-D-16-0014.1, 2017.

**Comments 2:** Line 43-46, actually, at low temperature (such as below -25 degree C), even sulfate and nitrate particles can serve as IN as indicated by Che et al. (2019, doi: 10.1016/j.atmosres.2020.105196).

Response 2: Thank you for providing this reference. We have added it to the revised manuscript.

**Comments 3:** *Line 55-57, this description is not correct in my opinion. The change of cirrus clouds here in principle reduces the outgoing longwave radiation to space while it increases the emission of cirrus, which is why it plays a more cooling effect.*

**Response 3:** For clarity, we have modified this sentence to **'These optically thinner cirrus** clouds absorb outgoing long-wave (LW) radiation from the surface and allow more LW radiation to emit to space, contributing to a cooling effect (Gasparini and Lohmann, 2016). This cooling effect prevails over the warming effect caused by the increased incoming solar radiation (warming), resulting in a net-positive radiative effect (cooling) on the radiation budget of the Earth (Kuebbeler et al., 2014; Lohmann and Gasparini, 2017).' (Please see lines 57-61)

**References:**

- Gasparini, B., and Lohmann, U.: Why cirrus cloud seeding cannot substantially cool the planet,J. Geophys. Res.-Atmos., 121, 4877–4893, doi.org/10.1002/2015JD024666. 2016.
- Lohmann. U., and Gasparini, B.: A cirrus cloud climate dial? Science, 357(6348), 248-249, doi.org/10.1126/science.aan3325, 2017.

Kuebbeler, M., Lohmann, U., Hendricks, J., and Kärcher, B.: Dust ice nuclei effects on cirrus

clouds, Atmos. Chem. Phys., 14, 3027-3046, doi.org/10.5194/acp-14-3027-2014, 2014.

**Comments 4:** *Line 89-90, considering potential variation of cirrus properties, why do the authors only choose two cases, and why do they choose these two cases?*

**Response 4:** Our manuscript is a technical report to introduce a sole remote-sensing approach to separate ice-nucleation regimes for cirrus clouds. Compared with in-situ techniques, our approach can realize long-term observation over the globe using space-borne platforms, such as CALIPSO and EarthCARE (Illingworth et al., 2015) satellite, which is of high significance to quantify relative contributions of homogeneous/heterogeneous freezing to cirrus clouds formation and evaluate the effects of different parameterization schemes on global cloud simulations. However, many aspects need to be verified for this method (such as the applicability of regional conversion factors in INP retrieval in section 2.3, the selection of optimal INP parameterization scheme in section 2.3, the confirmation of dust-related cirrus clouds in section 3, and so on), which is thus better to start with some clear-cut case studies. Then, a robust longterm study on a global scale can be expected in the future. That's why we submit this manuscript as the type of 'technical report'. Therefore, we have added the following sentence in the revised manuscript '..., which are favorable for validating some aspects of this method (such as the applicability of regional conversion factors in INP retrieval, the selection of optimal INP parameterization scheme, the confirmation of dust-related cirrus clouds, and so on) and conducting a robust long-term study on a global scale subsequently.' (Please see lines 94-97)

**Reference:**

Illingworth, A. J., Barker, H. W., Beljaars, A., Ceccaldi, M., Chepfer, H., Clerbaux, N., Cole, J., Delanoë, J., Domenech, C., Donovan, D. P., Fukuda, S., Hirakata, M., Hogan, R. J., Huenerbein, A., Kollias, P., Kubota, T., Nakajima, T., Nakajima, T. Y., Nishizawa, T., Ohno, Y., Okamoto, H., Oki, R., Sato, K., Satoh, M., Shephard, M., Velázquez-Blázquez, A., Wandinger, U.,Wehr, T., van Zadelhoff, G.-J.: The EarthCARE Satellite: The next step forward in global measurements of clouds, aerosols, precipitation and radiation, B. Am. Meteorol. Soc., 96, 1311–1332, doi.org/10.1175/BAMS-D-12-00227.1, 2015.

**Comments 5:** *Line 142-143, how could the authors make sure that the meteorology at the station is the same as that over the location CALIOP observes.*

**Response 5:** Generally, there are only two options for the meteorological parameters in INPC calculation, from either radiosonde measurements (He et al., 2021b) or the reanalysis data (ERA5 or GDAS) (Ansmann et al., 2019). We employ the radiosonde measurements since they have a better vertical resolution and accuracy for a fixed geo-location. As the reviewer mentioned, it contains an assumption that the atmosphere between the weather station and the location CALIPSO observes is horizontally homogeneous. This assumption is more valid for temperature and pressure (what we use here) than relative humidity. To be clear, we have added the following sentence 'It is assumed that the temperature and pressure profiles of the atmosphere between the weather station and the location CALIPSO observes are horizontally homogeneous.' (Please see lines 153-154)

**Comments 6:** *Line 144-148, for ice forming, only these two mechanisms play the role?**

**Response 6:** These two (immersion freezing and deposition freezing) are considered as the primary mechanism playing the role here. Besides, condensation freezing can be considered a special type of immersion freezing. Contacting freezing, which needs an INP to collide with a supercooled droplet is ignored (the description can be found in the text regarding to figure 6). We have added the sentence below 'Besides, condensation freezing can be considered a special type of immersion freezing; contacting freezing, which needs an INP to collide with a supercooled droplet, was ignored.' (Please see lines 160-162)

**Comments 7:** *Line 178, is there any method to confirm the interaction between dust and cirrus clouds?**

**Response 7:** The best method to confirm the dust-cirrus interaction is to sample the ice crystals inside cirrus clouds. Whether the residuals of sampled cloud particles contain dust particles can be a direct indicator of the interaction between dust and cirrus clouds (Cziczo et al., 2013). As for the sole employment of the remote sensing approach, it can be concluded the interaction between dust and cirrus clouds if cirrus clouds are observed to be **embedded** in the dust layer as also can be seen in figure 14 of Ansmann et al. (2019a). To better explain this issue, we have added the following statement '**The dust-cloud interaction is generally considered to take place if a cirrus cloud is embedded in a dust layer (Ansmann et al., 2019a; Marinou et al., 2019).' (Please see lines 191-192)**

Figure 14. Continuous cirrus and mixed-phase cloud observations for 30 h over Nicosia on 17–18 March 2015 (also shown in Fig. 5e, g, and i). The air mass from 5 to 10 km height was replaced (starting at great heights) by dust-free, dry air advected from Turkey and southern Europe between 02:00 and 11:00 UTC on 18 March, leading to the impression of a descending dust and cirrus layer. Several INPC and ICNC values estimated from the lidar observations are given as numbers determined for the indicated orange (INPC) and blue (ICNC) boxes. The deposition nucleation U17-I(d) parameterization is used on 17 March (at 9–10 km height for  $S_i = 1.1$ ) and the immersion freezing D15 parameterization is applied in the evening data analysis on 18 March (at 5–6 km height). Dashed white lines show the GDAS1 temperature isolines with a 3 h resolution.

**References:**

- Ansmann, A., Mamouri, R.-E., Bühl, J., Seifert, P., Engelmann, R., Hofer, J., Nisantzi, A., Atkinson, J. D., Kanji, Z. A., Sierau, B., Vrekoussis, M., and Sciare, J.: Ice-nucleating particle versus ice crystal number concentration in altocumulus and cirrus embedded in Saharan dust: A closure study, Atmos. Chem. Phys., 19, 15087-15115. doi.org/10.5194/acp-19-15087-2019, 2019a.
- Cziczo, D., Froyd, K., Hoose, C., Jensen, E., Diao, M., Zondlo, M., Smith, J., Twohy, C., and Murphy, D.: Clarifying the dominant sources and mechanisms of cirrus cloud formation, Science, 340, 1320-1324, doi.org/10.1126/science.1234145, 2013.
- Krämer, M., Rolf, C., Luebke, A., Afchine, A., Spelten, N., Costa, A., Meyer, J., Zöger, M., Smith, J., Herman, R. L., Buchholz, B., Ebert, V., Baumgardner, D., Borrmann, S., Klingebiel, M., and Avallone, L.: A microphysics guide to cirrus clouds – Part 1: Cirrus types, Atmos. Chem. Phys., 16, 3463-3483, doi.org/10.5194/acp-16-3463-2016, 2016.
- Marinou, E., Tesche, M., Nenes, A., Ansmann, A., Schrod, J., Mamali, D., Tsekeri, A., Pikridas, M., Baars, H., Engelmann, R., Voudouri, K.-A., Solomos, S., Sciare, J., Groß, S., Ewald, F., and Amiridis, V.: Retrieval of ice-nucleating particle concentrations from lidar observations and comparison with UAV in situ measurements, Atmos. Chem. Phys., 19, 11315-11342. doi.org/10.5194/acp-19-11315-2019, 2019.

**Comments 8:** Line 205-209, it seems that there is a large fraction of ice particles with diameters between 5 and 25  $\mu$ m (more than 50%). Is this reasonable, and why?

Response 8: The large fraction of ice crystals with diameters of 5-25 µm is common as seen in

the comparison with in-situs measurements (Sourdeval et al., 2018, see Figure 3 given as below and Appendix 3 therein). It is probably caused by the assumption of the ice-particle size spectrum in DARDAR Nice retrieval. Sourdeval et al. (2018) have the following statements in section 3.1 of their paper: '*However, these numbers do not provide a complete estimation of the accuracy of*  $N_i$  *as DARDAR does not rigorously account for uncertainties related to assumptions on the PSD shape. A preliminary sensitivity study has shown that strong deviations from the assumed a and*  $\beta$  *parameters could reasonably lead to errors of up to 50% on*  $N_i$  (see Sect. A3). Therefore, *the overall uncertainties on Ni due to instrumental sensitivity and physical assumptions are difficult to quantify based on DARDAR products alone.*'

Therefore, we have also added the following sentences in the outlook part of last section 'In addition, the future launch of the EarthCARE satellite is more anticipated (Illingworth et al., 2015), since its 94.05-GHz cloud profiling radar can possess the capability of Doppler detection so that the in-cloud ICNC will be determined more accurately under the better constraint of the ice-particle size spectrum.' (Please see lines 366-369)

---

## Author Comment (AC2)

**Response to reviewer #2**

**General Comments:**

**Content**

This manuscript describes an approach to determine the ice crystal number concentration (ICNC) and dust related ice nucleation particle concentration (INPC) from satellite and sun photometer observations. Both quantities are compared to each other to examine the nucleation pathway (homogeneous or heterogeneous) of two single cirrus cloud cases in China.

**Overall impression and rating**

The overall impression of the manuscript is good in general. The manuscript is mostly written in a clear way and the most important aspects are considered. The presentation quality of the manuscript is besides small details good. It is well organized and the analysis and results are clearly structured and communicated. However, in some minor parts I cannot fully follow the argumentation. Especially, the explanation of the middle cloud part in case 1 where INPC and ICNC disagree is too short and insufficient. For these reasons, I recommend publication in ACP after some minor manuscript revisions.

**Response:** We appreciate the reviewer for the thoughtful review and constructive comments, which are valuable to improving the quality of the manuscript. All the comments have been responded point to point as given below and the related modifications have been made in the revised manuscript.

**Specific comments:**

**Comments 1:** Page 2, lines 41-43, You are right that liquid origin clouds form completely heterogeneously, but at a later stage when reaching higher/colder altitudes additional homogeneous freezing can occur, if the updraft is fast enough. This can happen in convection and warm conveyor belts (WCB, see Kraemer et al. 2016). I recommend to add at the end of sentence: "...altitudes where homogeneous freezing can occur under high updraft conditions in addition to the heterogeneously formed ice crystals."

**Response 1:** According to the reviewer's suggestion, we have added the sentence '...where homogeneous freezing can occur under high updraft conditions in addition to the heterogeneously formed ice crystals.' (Please see lines 42-43)

**Comments 2:** Page 5, lines 137-138, What is actually meant by "interesting cirrus clouds"? Are this cases where your ICNC-INPC closure worked or how do you select the "interesting cases". Maybe you can a little bit more specific in the text.

**Response 2:** For the remote sensing approach, the interaction between dust and cirrus clouds can be concluded if cirrus clouds are observed to be embedded in dust layers as also can be seen in figure 14 of Ansmann et al. (2019a). Thus, we have added the following statement at the beginning of this paragraph 'The dust-cloud interaction is generally considered to take place if a cirrus cloud is embedded in a dust layer (Ansmann et al., 2019a; Marinou et al., 2019); in this case, the dust layer and cloud layer should have a spatial overlap either vertically or horizontally so that they can be considered as coupled.' (Please see lines 144-146) For clarity, we have also removed the word 'interesting'.

Figure 14. Continuous cirrus and mixed-phase cloud observations for 30 h over Nicosia on 17–18 March 2015 (also shown in Fig. 5e, g, and i). The air mass from 5 to 10 km height was replaced (starting at great heights) by dust-free, dry air advected from Turkey and southern Europe between 02:00 and 11:00 UTC on 18 March, leading to the impression of a descending dust and cirrus layer. Several INPC and ICNC values estimated from the lidar observations are given as numbers determined for the indicated orange (INPC) and blue (ICNC) boxes. The deposition nucleation U17-I(d) parameterization is used on 17 March (at 9–10 km height for  $S_i = 1.1$ ) and the immersion freezing D15 parameterization is applied in the evening data analysis on 18 March (at 5–6 km height). Dashed white lines show the GDAS1 temperature isolines with a 3 h resolution.

**References:**

- Ansmann, A., Mamouri, R.-E., Bühl, J., Seifert, P., Engelmann, R., Hofer, J., Nisantzi, A., Atkinson, J. D., Kanji, Z. A., Sierau, B., Vrekoussis, M., and Sciare, J.: Ice-nucleating particle versus ice crystal number concentration in altocumulus and cirrus embedded in Saharan dust: A closure study, Atmos. Chem. Phys., 19, 15087-15115. doi.org/10.5194/acp-19-15087-2019, 2019a.
- Marinou, E., Tesche, M., Nenes, A., Ansmann, A., Schrod, J., Mamali, D., Tsekeri, A., Pikridas, M., Baars, H., Engelmann, R., Voudouri, K.-A., Solomos, S., Sciare, J., Groß, S., Ewald, F., and Amiridis, V.: Retrieval of ice-nucleating particle concentrations from lidar observations and comparison with UAV in situ measurements, Atmos. Chem. Phys., 19, 11315-11342.

doi.org/10.5194/acp-19-11315-2019, 2019.

**Comments 3:** Page 5, line 144, I find this sentence confusing. What is meant by "isotherm" in this context? Cirrus clouds usually occur below -38 °C, but can also as completely frozen mixed-phase clouds above -38 °C. I recommend to rewrite this sentence to make it clearer.

**Response 3:** We have rewritten this sentence as **'The formation of cirrus clouds can be in situ-origin below -38 °C or liquid-origin from mixed-phase clouds above -38°C; thus, both...'** (Please see lines 155-156)

**Comments 4:** *Page 6, line 168, The date in the headline does not fit to the case you are showing! (29 December 2010 and 15 May 2008). So please correct. Please also capitalize after the colon.*

**Response 4:** We are sorry for the mistake. The date in the headline has been modified to '3.1 **Case on 15 May 2008: Sole presence of heterogeneous nucleation'.** (Please see line 181)

**Comments 5:** Page 6, line 173, I do not understand the first part of the sentence: "Dust particles were full of the cloud-free regions". Can you please rephrase it!

**Response 5:** We intend to mean the clouds formed within the dust layer. For clarity, we have removed this sentence from the manuscript.

**Comments 6:** Page 6, lines 182-184, How do you determine the average values? Just taking your bounding box shown in the figures or did you use the feature mask from CAPLISO? Because of the very irregular shape of the cloud, this can have a large influence on the averaged values. The same comment applies for page 8 line 238-239.

**Response 6:** For the first case, we have rechecked the calculation of average values. Now, all the data points within the altitude range of 9-11 km and latitude range of 33.2-35.0°N have been taken into calculation in order to be consistent with the calculations in Figures 5, 6, and Table 2. Although there are a series of irregular cirrus clouds, only identified cirrus clouds in the feature mask show the valid data for cloud extinction, effective radius, and ice water content (as seen in Figure 4), which are used in calculating the averaged values. Therefore, there is no significant

influence on the averaged values. These three averaged values have been slightly modified to '0.60 km-1, 34.93  $\mu$ m, and 13.89 mg m-3', respectively. Also, we have added the following sentence 'It should be mentioned that only the data points identified as cirrus clouds (with feature mask) and having valid data were used for calculating these averaging values.' (Please see lines 196-199)

For the second case, these three averaged values have been slightly modified to '0.47 km-1, 45.61  $\mu$ m, and 14.10 mg m-3', respectively. Here we have taken all the data points within the altitude range of 5-10 km and latitude range of 33-35°N into the calculation. Only data points identified as cirrus clouds (in the feature mask) and having valid data were used for calculating the averaged values. We have added the following sentence 'Note that only the data points identified as cirrus clouds (with feature mask) and having valid data were used for calculating these averaging values.' (Please see lines 259-262)

**Comments 7:** Page 8, lines 221-225, The explanation and discussion of the disagreement in the lower part of the cloud is definitely not sufficient and long enough. You argue with sedimentation of the heterogeneously formed ice crystals from the cloud top level, which can be an explanation. But if the heterogeneously formed ice crystals fall from above they also lower the ice concentration and also the INPC which are consumed by the formation process in the upper part of the cloud. In addition, new ice crystals in the upper part cannot form heterogeneously afterwards anymore because of low INPC values. Or you have to explain where new IN particles should come from. I also do not understand the argument with homogeneous freezing in the lower part of the cloud. Homogenous freezing would need higher vertical updrafts to maintain high supersaturations. Where should the higher vertical updrafts come from and why one could not find them in the top part of the cloud? I also think that homogeneous freezing can still not be ruled out completely also for the upper part of the cloud especially with the argument of ice crystal sedimentation. I think this point should be discussed in more detail in this Section of the paper.

**Response 7:** Thanks for pointing out the inappropriate discussions on the large ICNC below and the small ICNC above in this case. Indeed, it is rather hard to provide a process-level description of the cirrus clouds solely with space-borne **snapshot observation**. We can only speculate on the possible process underlying based on the information on cloud geometric shape, in-cloud ICNC, and dust-related INPC in the vicinity. Nevertheless, these remote sensing observations

and values obtained by advanced aerosol/cloud retrievals provide unique information to interpret cloud processes and promote our understanding of aerosol-cloud interactions at a global scale. To make readers bear this disadvantage in mind, we have rephrased these sentences as follows 'These large ICNCs are possibly attributed to the occurrence of homogeneous nucleation. Consequently, both heterogeneous and homogeneous nucleation might take place in this case. Without airborne in-situ observations, the process-level evolution of these cirrus clouds cannot be well described since space-borne active observations only provide snapshot information of clouds. As seen from the geometric shape (small horizontal coverage and large vertical extent) of cirrus clouds, they were likely to form via homogeneous nucleation first accompanied by a fast updraft condition at lower altitudes, causing the large ICNCs at below. In this type of cirrus cloud, sedimentation of ice crystals is considered not to play an important role. Then, along with the updraft, water vapor was consumed gradually and the in-cloud RHi would quickly reduce to close to saturation; thus, heterogeneous nucleation would take charge predominantly at higher altitudes (as discussed for the lower part of cirrus clouds in the last paragraph) (Krämer et al., 2016, **2020).**' (Please see lines 236-245)

Also, we have added some sentences to discuss the potential benefits of involving the groundbased observations in the last paragraph of manuscript. 'CALIOP level-2 data product with the 5-km horizontal resolution cannot satisfy the accurate identification of dust layer and cirrus cloud on a small scale (Vaillant de Guélis et al. 2022), causing a potential to overestimate dust-related INPC, which can be solved by ground-based lidar observations with higher spatio-temporal resolution. With ground-based observations, the involved measurements of the Doppler velocity of ice crystals and the vertical velocity of airflows will be more beneficial to determine the accurate ICNC and the process-level characterization of cirrus formation (Bühl et al., 2015, 2016, 2019; Radenz et al., 2018, 2021). In addition, the future launch of the EarthCARE satellite can promote our understanding of cloud processes (Illingworth et al., 2015), since its 94.05-GHz cloud profiling radar can possess the capability of Doppler detection so that the in-cloud ICNC will be determined more accurately under the better constraint of the ice-particle size spectrum.' (Please see lines 361-369)

**References:**

Bühl, J., Leinweber, R., Görsdorf, U., Radenz, M., Ansmann, A., and Lehmann, V.: Combined vertical-velocity observations with Doppler lidar, cloud radar and wind profiler, Atmos.

Meas. Tech., 8, 3527–3536, https://doi.org/10.5194/amt-8-3527-2015, 2015.

- Bühl, J., Seifert, P., Myagkov, A., and Ansmann, A.: Measuring ice- and liquid-water properties in mixed-phase cloud layers at the Leipzig Cloudnet station, Atmos. Chem. Phys., 16, 10609–10620, https://doi.org/10.5194/acp-16-10609-2016, 2016.
- Bühl, J., Seifert, P., Radenz, M., Baars, H., and Ansmann, A.: Ice crystal number concentration from lidar, cloud radar and radar wind profiler measurements, Atmos. Meas. Tech., 12, 6601–6617, doi.org/10.5194/amt-12-6601-2019, 2019.
- Krämer, M., Rolf, C., Luebke, A., Afchine, A., Spelten, N., Costa, A., Meyer, J., Zöger, M., Smith, J., Herman, R. L., Buchholz, B., Ebert, V., Baumgardner, D., Borrmann, S., Klingebiel, M., and Avallone, L.: A microphysics guide to cirrus clouds – Part 1: Cirrus types, Atmos. Chem. Phys., 16, 3463-3483, doi.org/10.5194/acp-16-3463-2016, 2016.
- Krämer, M. Rolf, C., Spelten, N., Afchine, A., Fahey, D., Jensen, E., Khaykin, S., Kuhn, T., Lawson, P., Lykov, A., Pan, L., Riese, M., Rollins, A., Stroh, F., Thornberry, T., Wolf, V., Woods, S., Spichtinger, P., Quaas, J., and Sourdeval, O.: A microphysics guide to cirrus—part 2: climatologies of clouds and humidity from observations, Atmos. Chem. Phys. 20, 12569–12608, doi.org/10.5194/acp-20-12569-2020, 2020.
- Radenz, M., Bühl, J., Lehmann, V., Görsdorf, U., and Leinweber, R.: Combining cloud radar and radar wind profiler for a value added estimate of vertical air motion and particle terminal velocity within clouds, Atmos. Meas. Tech., 11, 5925–5940, https://doi.org/10.5194/amt-11-5925-2018, 2018.
- Radenz, M., Bühl, J., Seifert, P., Baars, H., Engelmann, R., Barja González, B., Mamouri, R. E., Zamorano, F., and Ansmann, A.: Hemispheric contrasts in ice formation in stratiform mixed-phase clouds: disentangling the role of aerosol and dynamics with ground-based remote sensing, Atmos. Chem. Phys., 21(23), 17969-17994, doi.org/10.5194/acp-21-17969-2021, 2021.
- Vaillant de Guélis, T., Ancellet, G., Garnier, A., C.-Labonnote, L., Pelon, J., Vaughan, M. A., Liu, Z., and Winker, D. M.: Assessing the benefits of Imaging Infrared Radiometer observations for the CALIOP version 4 cloud and aerosol discrimination algorithm, Atmos. Meas. Tech., 15, 1931–1956, https://doi.org/10.5194/amt-15-1931-2022, 2022.
- Illingworth, A. J., Barker, H. W., Beljaars, A., Ceccaldi, M., Chepfer, H., Clerbaux, N., Cole, J., Delanoë, J., Domenech, C., Donovan, D. P., Fukuda, S., Hirakata, M., Hogan, R. J.,

Huenerbein, A., Kollias, P., Kubota, T., Nakajima, T., Nakajima, T. Y., Nishizawa, T., Ohno, Y., Okamoto, H., Oki, R., Sato, K., Satoh, M., Shephard, M., Velázquez-Blázquez, A., Wandinger, U., Wehr, T., van Zadelhoff, G.-J.: The EarthCARE Satellite: The next step forward in global measurements of clouds, aerosols, precipitation and radiation, B. Am. Meteorol. Soc., 96, 1311–1332, doi.org/10.1175/BAMS-D-12-00227.1, 2015.

**Comments 8:** Page 9, lines 277-279, I cannot follow your conclusion that "heterogeneous nucleation would gradually be dominant" in a competition situation. When you uplift an air parcel you will increase relative humidity until a cloud is forming. Of course, you would form a cloud heterogeneously first, but with further vertical updraft and thus cooling, relative humidity can increase again reaching homogeneous nucleation threshold even if you are consuming humidity by particle growth of heterogeneously formed ice crystals. Thus, forming heterogeneously and homogeneously exactly at the same time is not possible, but of course one after the other. And then I would identify the dominance of formation mechanism by the ICNC. Given the high ICNC values in the upper part of the cloud, I would argue that you have a large dominance of homogeneous frozen ice crystals in the cirrus cloud. I suggest to rethink your conclusions and discussion in this point.

**Response 8:** We are grateful for pointing out this issue. We have rephrased the discussion of this part as follows 'Therefore, we can conclude that both heterogeneous and homogeneous nucleation had taken place during the formation of this cirrus cloud. (...) Since the observation of vertical velocity was lacking, it is hard to determine the exact process of cirrus formation. In this case, it is likely that the cirrus cloud first formed via heterogeneous nucleation under a slow updraft condition and further switched to a 'second stage' in which homogeneous nucleation began to be dominant owing to the persistence of cooling/uplifting (Krämer et al., 2016, 2020). Krämer et al., (2016) mentioned that this type of cirrus usually has a large geographic coverage, which can also be seen in this case. Considering the large ICNC, homogeneous nucleation should be the dominant type of ice nucleation.' (Please see lines 304-314)

**Reference:**

Krämer, M., Rolf, C., Luebke, A., Afchine, A., Spelten, N., Costa, A., Meyer, J., Zöger, M., Smith, J., Herman, R. L., Buchholz, B., Ebert, V., Baumgardner, D., Borrmann, S., Klingebiel, M., and Avallone, L.: A microphysics guide to cirrus clouds – Part 1: Cirrus types, Atmos. Chem. Phys., 16, 3463-3483, doi.org/10.5194/acp-16-3463-2016, 2016.

Krämer, M. Rolf, C., Spelten, N., Afchine, A., Fahey, D., Jensen, E., Khaykin, S., Kuhn, T., Lawson, P., Lykov, A., Pan, L., Riese, M., Rollins, A., Stroh, F., Thornberry, T., Wolf, V., Woods, S., Spichtinger, P., Quaas, J., and Sourdeval, O.: A microphysics guide to cirrus—part 2: climatologies of clouds and humidity from observations, Atmos. Chem. Phys. 20, 12569–12608, doi.org/10.5194/acp-20-12569-2020, 2020.

**Comments 9:** Figure 12: In part (a) of the figure you write T<-38°C, while you write <0°C in the caption. In this case your argumentation about thin clouds (few large crystals) is only valid for in-situ cirrus clouds, I suggest to change the caption to T<-38°C as in the Figure. In part (b) the RHi values stated in the figure are not correct. In a competition case heterogeneous freezing still happens at RHi values 100-140%. Thus, I suggest to change the values in the Figure and Caption to 100-170%.

**Response 9:** In the caption, 'temperature of  $< 0^{\circ}$ C' is a mistake. Thanks for pointing out this mistake. We have modified ' $<0^{\circ}$ C' to ' $<-38^{\circ}$ C'. As for figure 12b, we intend to limit the RHi values to '140-170%' for the situations in which both homogeneous and heterogeneous freezing occur. Therefore, although heterogeneous freezing still occurs at RHi values of 100-140%, we consider that, if possible, it would be reasonable to retain the '140-170%'.

**Technical comments:**

Comments 1: Page 1, line 27: Better write "~5km up to the tropopause".

**Response 1:** 'from  $\sim$ 5 km to up to the tropopause' has been modified to ' $\sim$ 5 km up to the tropopause'.

Comments 2: Page 2, line 32: Use the plural form "general circulation models".

Response 2: 'model' has been replaced by 'models'.

**Comments 3:** Page 3, line 84: I recommend to include Kraemer et al 2020 as reference in addition to Marinou et al. 2019. The authors also perform a comparison between in-situ and

satellite ICNC.

**Response 3:** Kraemer et al., 2020 has been added as a reference here.

**Comments 4:** *Page 6, line 175: Please change "can be considered dust-related cirrus clouds" to "can be considered as dust-related cirrus clouds".*

**Response 4:** 'as' has been added.

**Comments 5:** *Page 7, line 199: "above -35 °C isotherm". I guess you mean colder than -35 °C? So please correct the wording.*

Response 5: 'above -35 °C isotherm' has been modified to 'At temperatures colder than -35 °C'.

**Comments 6:** Page 7, lines 200-204, Table 2: Please explain what is meant by the parameter Si, because it is not mentioned in the text before or afterwards.

**Response 6:** Thank you for pointing out this.  $S_i$  denotes the ice saturation ratio. We have added its definition in the context here as well as in the captions of Tables 1 and 2 and Figures 6 and 11.

**Comments 7:** *Page 8, line 221: "ice crystals falling": Better use the common term "ice crystal sedimentation".*

Response 7: 'falling' has been replaced by 'sedimentation'.

Comments 8: Page 8, line 226: Please capitalize after the colon.

Response 8: 'competition' has been modified to 'Competition'.

**Comments 9:** Figure 2/3/4: I would recommend to zoom a little bit closer to your selected cirrus case by showing only date between e.g. 45-25° latitude. Then all features are better visible.

Response 9: We are grateful for the reviewer's suggestion. The latitude range of 25-45° is shown

in updated Figures 2-4. The features are now better visible.

**Comments 10:** *Figure 3/4: Please use the same latitudinal projection as in Figure 2 to make the figures better comparable.*

Response 10: The latitudinal projections are uniform for revised Figures 2-4.

**Comments 11:** *Figure 8/9: Same comment for Figure 3/4 above.*

**Response 11:** Similar to Figures 2-4, the latitude range of 25-45° is shown in updated Figures 7-9. Besides, the latitudinal projections are uniform for revised Figures 7-9.